# DESCRIBE-AND-DISSECT: INTERPRETING NEURONS IN VISION NETWORKS WITH LANGUAGE MODELS

## ABSTRACT

In this paper, we propose Describe-and-Dissect, a novel method to describe the roles of hidden neurons in vision networks. Describe-and-Dissect utilizes recent advancements in multimodal deep learning to produce complex natural language descriptions, without the need for labeled training data or a predefined set of concepts to choose from. Additionally, Describe-and-Dissect is *training-free*, meaning we don't train any new models and can easily leverage more capable general purpose models in the future. We show on a large scale user study that our method outperforms the state-of-the-art baseline methods including CLIP-Dissect, MILAN, and Network Dissection. Our method on average provides the highest quality labels and is more than $2\times$ as likely to be selected as the best explanation for a neuron than the best baseline.

## 1 INTRODUCTION

Recent advancements in Deep Neural Networks (DNNs) within machine learning have enabled unparalleled development in multimodal artificial intelligence. While these models have revolutionized domains across image recognition, natural language processing, and automation, they haven't seen much use in various safety-critical applications, such as healthcare or ethical decision-making. This is in part due to their cryptic "black box" nature, where the internal workings of complex neural networks have remained beyond human comprehension. This makes it hard to place appropriate trust in the models and additional insight in their workings is needed to reach wider adoption.

Previous methods have gained a deeper understanding of DNNs by examining the functionality (we also refer to as *concepts*) of individual neurons[1]. This includes works based on manual inspection Erhan et al. (2009); Zhou et al. (2014); Olah et al. (2020); Goh et al. (2021), which can provide high quality description at the cost of being very labor intensive. Alternatively, Network Dissection (Bau et al., 2017) automated this labeling process by creating the pixelwise labeled dataset, *Broden*, where fixed concept set labels serve as ground truth binary masks for corresponding image pixels. The dataset was then used to match neurons to a label from the concept set based on how similar their activation patterns and the concept maps were. While earlier works, such as Network Dissection, were restricted to an annotated dataset and a predetermined concept set, CLIP-Dissect (Oikarinen & Weng, 2023) offered a solution by no longer requiring labeled concept data, but still requires a predetermined concept set as input. By utilizing OpenAI's CLIP model, CLIP-Dissect matches neurons to concepts based on their activations in response to images, allowing for a more flexible probing dataset and concept set compared to previous works.

However, a major limitation still arises from these methods: Concepts detected by certain neurons, especially in intermediate layers, prove to be difficult to encapsulate using the simple, often single-word descriptions provided in a fixed concept set. MILAN (Hernandez et al., 2022) sought to enhance the quality of these neuron labels by providing generative descriptions, but their method requires training a new descriptions model from scratch to match human explanation on a dataset of neurons. This leads to their proposed method being more brittle and often performs poorly outside its training data.

To overcome these limitations, we propose Describe-and-Dissect (abbreviated as **DnD**), a pipeline to *dissect* DNN by utilizing an image-to-text model to *describe* highly activating images for cor-

---

[1]We conform to prior works' notation and use "neuron" to describe a channel in CNNs.

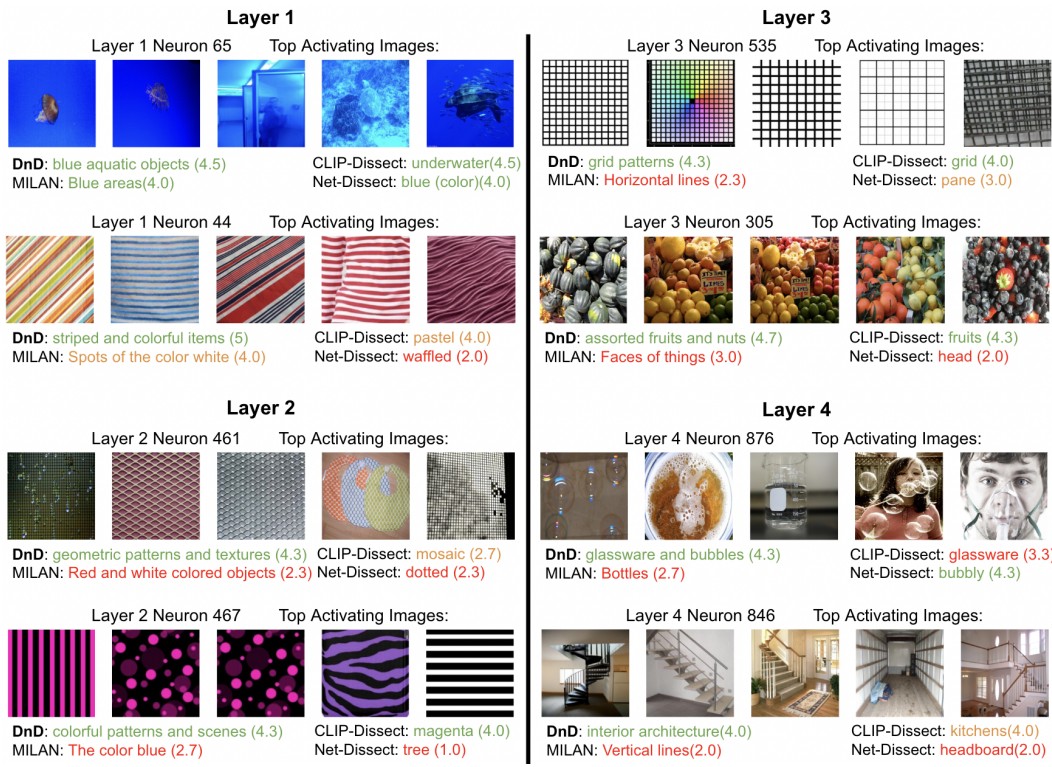

Figure 1: Neuron descriptions provided by our method (**DnD**) and baselines CLIP-Dissect Oikarinen & Weng (2023), MILAN Hernandez et al. (2022)), and Network Dissection Bau et al. (2017) for random neurons from ResNet-50 trained on ImageNet. We have added the average quality rating from our Amazon Mechanical Turk experiment described in section 4.1 next to each label and color-coded the neuron descriptions by whether we believed they were accurate, somewhat correct or vague/imprecise.

responding neurons. The descriptions are then semantically combined by a large language model, and finally refined with synthetic images to generate the final concept of a neuron. We show that Describe-and-Dissect provides more complex and higher-quality descriptions (up to 2-4× better) of intermediate layer neurons than other contemporary methods in a large scale user study. Example descriptions from our method are displayed in Figure 1and Appendix B.11 .

## 2 BACKGROUND AND RELATED WORK

### 2.1 NEURON INTERPRETABILITY METHODS

Network Dissection (Bau et al., 2017) is the first method developed to automatically describe individual neurons' functionalities. The authors first defined the densely-annotated dataset *Broden*, denoted as $\mathcal{D}_{\text{Broden}}$, as a ground-truth concept mask. The dataset is composed of various images $x_i$, each labeled with concepts $c$ at the pixel-level. This forms a ground truth binary mask $L_c(x_i)$ which is used to calculate the intersection over union (IoU) score between $L_c(x_i)$ and the binary mask from the activations of the neuron $k$ over all images $x_i \in \mathcal{D}_{\text{Broden}}$, denoted $M_k(x_i)$: $\text{IoU}_{k,c} = \frac{\sum_{x_i \in \mathcal{D}_{\text{Broden}}} M_k(x_i) \cap L_c(x_i)}{\sum_{x_i \in \mathcal{D}_{\text{Broden}}} M_k(x_i) \cup L_c(x_i)}$. The concept $c$ is assigned to a neuron $k$ if $\text{IoU}_{k,c} > \eta$, where the threshold $\eta$ was set to 0.04. Intuitively, this method finds the labeled concept whose presence in the image is most closely correlated with the neuron having high activation. Extensions of Network Dissection were proposed by Bau et al. (2020) and Mu & Andreas (2020).

However, Network Dissection is limited by the need of concept annotation and the concept set is a closed set that may be hard to expand. To address these limitations, a recent work CLIP-Dissect

Table 1: Comparison of properties of existing automated neuron labeling methods and Describe-and-Dissect. Green and boldfaced **Yes** or **No** indicates the desired property for a column.

| Method \ property | Requires Concept Annotations | Training Free | Generative Natural Language Descriptions | Uses Spatial Activation Information | Can easily leverage better future models |
|---|---|---|---|---|---|
| Network Dissection | Yes | **Yes** | No | **Yes** | No |
| MILAN | Training only | No | **Yes** | **Yes** | No |
| CLIP-Dissect | **No** | **Yes** | No | No | **Yes** |
| FALCON | **No** | **Yes** | No | **Yes** | **Yes** |
| **DnD (Ours)** | **No** | **Yes** | **Yes** | **Yes** | **Yes** |

(Oikarinen & Weng, 2023) utilizes OpenAI's multimodal CLIP (Radford et al., 2021) model to describe neurons automatically without requiring annotated concept data. They leverage CLIP to score how similar each image in the probing dataset $\mathcal{D}_{probe}$ is to the concepts in a user-specified concept set, to generate a concept activation matrix. To describe a neuron, they compare the activation pattern of said neuron to activations of different concepts on the probing data, and find the concept that is the closest match using a similarity function, such as softWPMI. Another very recent work FALCON (Kalibhat et al., 2023) uses a method similar to CLIP-dissect but augments it via counterfactual images by finding inputs similar to highly activating images with low activation for the target neuron, and utilizing spatial information of activations via cropping. However, they solely rely on cropping the most salient regions within an probing image to filter spurious concepts that are loosely related to the ground truth functionality labels of neurons. This approach largely restrict their method to local concept while overlooking holistic concepts within images, as also noted in (Kalibhat et al., 2023). Their approach is also limited to single word / set of words description that is unable to reach the complexity of natural language.

MILAN (Hernandez et al., 2022) is a different approach to describe neurons using natural language descriptions in a generative fashion. Note that despite the concept set in CLIP-Dissect and FALCON are flexible and open, they cannot provide generative natural language descriptions like MILAN. The central idea of MILAN is to train an images-to-text model from scratch to describe the neurons role based on 15 most highly activating images. Specifically, it was trained on crowdsourced descriptions for 20,000 neurons from select networks. MILAN can then generate natural language descriptions to new neurons by outputting descriptions that maximize the weighted pointwise mutual information (WPMI) between the description and the active image regions. One major limitation of MILAN is that the method require training a model to imitate human descriptions of image regions on relatively small training dataset, which may cause inconsistency and poor explanations further from training data. In contrast, our Describe-and-Dissect is *training-free*, *generative*, and produces a *higher quality of neuron descriptions* as supported by our extensive experiments in Figure 1, Table 2, and Table 3. A detailed comparison between our method and the baseline methods is shown in Table 1.

## 2.2 LEVERAGING LARGE PRETRAINED MODELS

In our **DnD** pipeline, we are able to leverage recent advances in the large pre-trained models to provide high quality and generative neuron descriptions for DNNs in a *training-free* manner. Below we briefly introduce the Image-to-Text Model, Large Language Models and Text-to-Image Model used in our pipeline implementation. The first model is Bootstrapping Language-Image Pretraining (BLIP) (Li et al., 2022), which is an image-to-text model for vision-language tasks that generates synthetic captions and filters noisy ones, employing bootstrapping for the captions to utilize noisy web data. While our method can use any image-to-text model, we use BLIP in this paper for our step 2 in the pipeline due to BLIP's high performance, speed, and relatively low computational cost. Note that better image captioning models such as BLIP2 can potentially increase the performance of our method, but were not used for our needs due to computational efficiency and cost.

The second model is GPT-3.5 Turbo, which is a transformer model developed by OpenAI for understanding and generating natural language. It provides increased performance from other contemporary models due to its vast training dataset and immense network size. We utilize GPT-3.5 Turbo

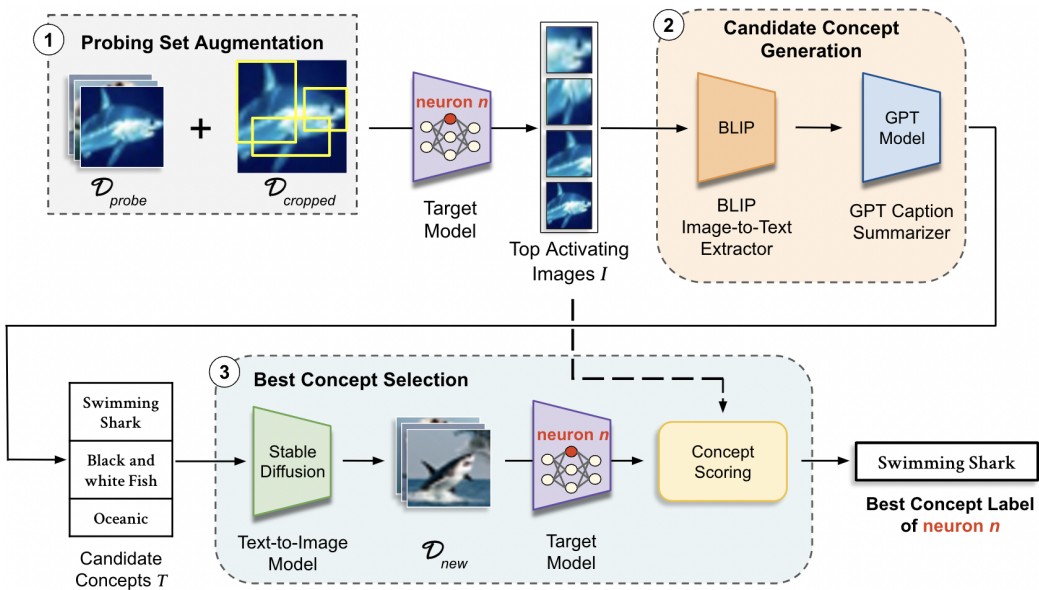

Figure 2: Overview of Describe-and-Dissect (**DnD**) algorithm.

for natural language processing and semantic summarization in the step 2 of our **DnD**. Note that we use GPT-3.5 Turbo in this work as it's one of the SOTAs in LLMs and cheap to use, but our method is compatible with other future and more advanced LLMs.

The third model is Stable Diffusion (Rombach et al., 2022), which is a text-to-image latent diffusion model (LDM) trained on a subset from the LAION-5B database (Schuhmann et al., 2022). By performing the diffusion process over the low dimensional latent space, Stable Diffusion is significantly more computationally efficient than other diffusion models. Due to its open availability, lower computational cost, and high performance, we employ Stable Diffusion for our image generation needs in the step 3 of **DnD**.

## 3 DESCRIBE-AND-DISSECT: METHODS

**Overview.** In this section, we present Describe-and-Dissect (**DnD**), a comprehensive method to produce generative neuron descriptions in deep vision networks. Our method is training-free, model-agnostic, and can be easily adapted to utilize advancements in multimodal deep learning. **DnD** consists of three steps:

- **Step 1. Probing Set Augmentation:** Augment the probing dataset with attention cropping to include both global and local concepts;
- **Step 2. Candidate Concept Generation:** Generate initial concepts by describing highly activating images and subsequently summarize them into candidate concepts using GPT;
- **Step 3. Best Concept Selection:** Generate new images based on candidate concepts and select the best concept based on neuron activations on these synthetic images with a scoring function.

An overview of Describe-and-Dissect (**DnD**) and these 3 steps is illustrated in Fig. 2.

### 3.1 STEP 1: PROBING SET AUGMENTATIONS

Probing dataset $\mathcal{D}_{probe}$ is the set of of images we record neuron activations on before generating a description. As described in section 2.1, one major limitation of (Kalibhat et al., 2023) is the restriction to local concepts while overlooking holistic concepts within images, while one limitation of (Oikarinen & Weng, 2023) is not incorporating the spatial activation information. Motivated by

these limitations, **DnD** resolves these problems by *augmenting* the original probing dataset with a set of attention crops of the highest activating images from the original probing dataset. The attention crops can capture the spatial information of the activations and we name this set as $\mathcal{D}_{cropped}$, shown in Fig. 2. To construct $\mathcal{D}_{cropped}$, **DnD** uses Otsu's method (Otsu, 1979) to generate a binary masked activation map, $M_n(x_i)$, by computing the optimal global threshold, $\lambda$, that maximizes interclass variance in $A_n(x_i)$, where $A_n$ is the activation map of neuron $n$ on the input $x_i$. **DnD** then performs attention cropping on bounding boxes derived from contours of the most salient regions in $M_n(x_i)$ using OpenCV (Bradski, 2000). Since thresholding filters counterfactual regions in $x_i$, we select $\alpha = 4$ largest bounding box regions within $M_n(x_i)$ that have a Intersection over Union(IoU) score less than $\eta = 0.5$ with all other previously cropped regions. Once $\mathcal{D}_{cropped}$ is formed, we use it along with $\mathcal{D}_{probe}$ as the augmented probing dataset to probe the target model.

### 3.2 STEP 2: CANDIDATE CONCEPT GENERATION

The top $K = 10$ activating images for a neuron $n$ are collected in set $I, |I| = K$, by selecting $K$ images $x_i \in \mathcal{D}_{probe} \cup \mathcal{D}_{cropped}$ with the largest $g(A_k(x_i))$. Here $g$ is a summary function (for the purposes of our experiments we define $g$ as the spatial mean) and $A_k(x_i)$ is the activation map of neuron $k$ on input $x_i$. We then generate a set of candidate concepts for the neuron with the following two part process:

- **Step 2A - Generate descriptions for highly activating images:** We utilize BLIP image-to-text model to generatively produce an image caption for each image in $I$. For an image $I_{j \in [K]}$, we feed $I_j$ into the base BLIP model to obtain an image caption.
- **Step 2B - Summarize similarities between image descriptions:** Next we utilize OpenAI's GPT-3.5 Turbo model to summarize similarities between the $K$ image captions for each neuron being checked. GPT is prompted to generate $N$ descriptions which identify and summarize the conceptual similarities between most of the BLIP-generated captions.

The output of **Step 2B** is a set of $N$ descriptions which we call "candidate concepts". We denote this set as $T = \{T_1, ..., T_N\}$. For the purposes of our experiments, we generate $N = 5$ candidate concepts unless otherwise mentioned. The exact prompt used for GPT summarization is shown in Appendix A.2.

### 3.3 STEP 3: BEST CONCEPT SELECTION

The last crucial component of **DnD** is concept selection, which selects the concept from the set of candidate concepts $T$ that is most correlated to the activating images of a neuron. We first use the Stable Diffusion model (Rombach et al., 2022) from Hugging Face to generate images for each concept $T_{j \in [N]}$. Generating new images is important as it allows us to differentiate between neurons truly detecting a concept or just spurious correlations in the probing data. The resulting set of images is then fed through the target model again to record the activations of a target neuron on the new images. Finally, the candidate concepts are ranked using a concept scoring model, as discussed below in section 3.4.

**Concept Selection Algorithm**   It consists of 4 substeps. For each neuron $n$, we start by:

1. *Generate supplementary images.* Generate $Q$ synthetic images using a text-to-image model for each label $T_{j \in [N]}$. The set of images from each concept is denoted as $\mathcal{D}_j, |\mathcal{D}_j| = Q$. The total new dataset is then $\mathcal{D}_{new} = \bigcup_{j=1}^{N} \mathcal{D}_j = \{x_1^{new}, ..., x_{N \cdot Q}^{new}\}$, which represents the full set of generated images. For the purposes of the experiments in this paper, we set $Q = 10$.

2. *Feed new dataset $\mathcal{D}_{new}$, back into the target model and rank the images based on activation.* We then evaluate the activations of target neuron $n$ on images in $\mathcal{D}_{new}$ and compute the rank of each image in terms of target neuron activation. Given neuron activations $A_n(x_i^{new})$, we define $\mathcal{G}_n = \{g(A_n(x_1^{new})), ..., g(A_n(x_{N \cdot Q}^{new}))\}$ as the set of scalar neuron activations.

3. *Gather the ranks of images corresponding to concept $T_j$.* Let Rank$(x; \mathcal{G})$ be a function that returns the rank of an element $x$ in set $\mathcal{G}$, such that Rank$(x'; \mathcal{G}) = 1$ if $x'$ is the

largest element in $\mathcal{G}$. For every concept $T_j$, we record the ranks of images generated from the concept in $\mathcal{H}_j$, where $\mathcal{H}_j = \{\text{Rank}(g(A_n(x)); \mathcal{G}_n) \ \forall \ x \in \mathcal{D}_j\}$, and $\mathcal{H}_j$ is sorted in increasing order, so $\mathcal{H}_{j1}$ is the rank of the lowest ranking element.

4. *Assign scores to each concept.* The scoring function $score(\mathcal{H}_j)$ assigns a score to a concept using the rankings of the concept's generated images, and potential additional information. The concept with the best (highest) score in $T$ is selected as the concept label for the neuron. Concept scoring functions are discussed below in section 3.4.

While we only experiment with Best Concept selection within the **DnD** framework, it can be independently applied with other methods like (Bau et al., 2017; Hernandez et al., 2022; Oikarinen & Weng, 2023) to select the best concept out of their top-k best descriptions, which is another benefit of our proposed method.

## 3.4 SCORING FUNCTION

Choosing the proper scoring function is very important to the accuracy of Concept Selection. Simplest scoring functions, such as mean, can be easily skewed by outliers in $\mathcal{H}_j$, resulting in final concepts that are loosely related to the features detected by neurons. In this section, we discuss 3 different scoring functions and a combination of them which we experimented with in section 4.2. The concept with highest score is chosen as the final description for neuron $n$.

- *Mean.* Simply score concept $j$ as the negative mean of its image activation rankings $\mathcal{H}_j$. Concepts where each image activates the neuron highly will receive low ranks and therefore high score. We use the subscript $M$ to denote it's the score using *Mean*.

$$score_M(\mathcal{H}_j) = -\frac{1}{Q}\sum_{i=1}^{Q}\mathcal{H}_{ji}$$

- *TopK Squared.* Given $\mathcal{H}_j$, the score is computed as the mean of the squares of $\beta$ lowest ranking images for the concept. For our experiments, we use $\beta = 5$. This is reduces reliance on few poor images. We use the subscript $TK$ to denote it's the score using *TopK Squared*:

$$score_{TK}(\mathcal{H}_j, \beta) = -\frac{1}{\beta}\sum_{i=1}^{\beta}\mathcal{H}_{ji}^2$$

- *Image Products.* Let set $\mathcal{D}_j^t \subset \mathcal{D}_j$, such that it keeps the $t$ highest activating images from $\mathcal{D}_j$. From the original set of activating images $I$ and $\mathcal{D}_j^t$, the Image Product score is defined as the average cosine similarity between original highly activating images and the generated images for the concept $j$. We measure this similarity using CLIP-ViT-B/16 Radford et al. (2021) as our image encoder $E(\cdot)$:

$$score_{IP}(I, \mathcal{D}_j^t) = \frac{1}{|I| \cdot |\mathcal{D}_j^t|}\sum_{x \in I}\sum_{x^{new} \in \mathcal{D}_j^t}(E(x) \cdot E(x^{new}))$$

See figure 3 for an illustration of Image Product. Intuitively, Image Products selects the candidate concept whose generated images are most similar to the original highly activating images. However, Image Product doesn't really account for how highly the new images actually activate the target neuron, which is why we chose to use this method to supplement other scoring functions. The enhanced scoring method, *TopK Squared + Image Products*, is described below.

- *TopK Squared + Image Products.* This scoring function uses both image similarity and neuron activations to select the best concept. The method combines Image Products and TopK Squared by multiplying the Image Product score with the relative rankings of each concept's TopK Squared score. We define $\mathcal{R}_{TK} = \{ \ score_{TK}(\mathcal{H}_j, \beta), \ \forall \ j \in \{1, ..., N\}\}$ as the set of TopK Squared scores for different descriptions. The final score is then:

$$score_{TK-IP}(\mathcal{H}_j, \beta, I, \mathcal{D}_j^t) = (N - \text{Rank}(score_{TK}(\mathcal{H}_j, \beta); \mathcal{R}_{TK})) \cdot score_{IP}(I, \mathcal{D}_j^t),$$

where we use $N - \text{Rank}(\cdot)$ to invert the ranks of *TopK Squared* so low ranks result in a high score.

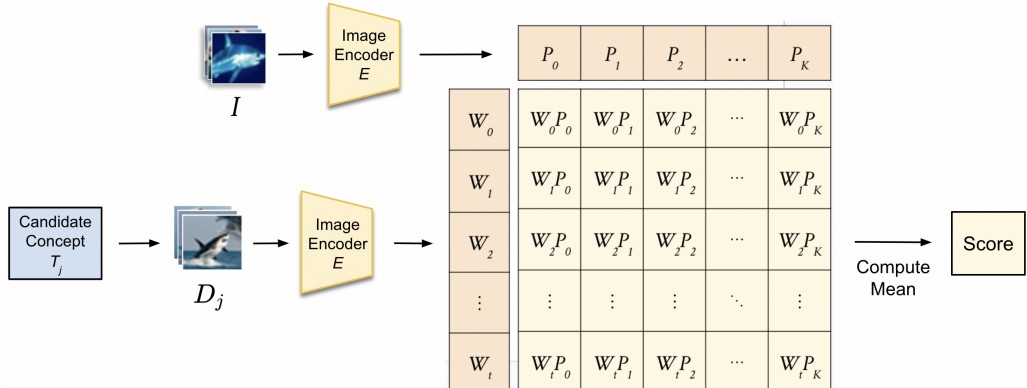

Figure 3: **Image Product Scoring Function.** In the diagram, we let $W_i = E(\mathcal{D}_j^i)$ and $P_i = E(I_i)$. Image Products computes the mean of $W_i \cdot P_i$.

In section 4.2, we compare the different functions, and find *TopK Squared + Image Product* to be the best for our purposes. We use this scoring function for all experiments unless otherwise specified.

# 4 EXPERIMENT

In this section we run our algorithm to describe the hidden neurons of two networks: ResNet-50 He et al. (2016) trained on ImageNet Russakovsky et al. (2015), and ResNet-18 He et al. (2016) trained on Places365 Zhou et al. (2016). We mostly relied on human evaluations to rate the quality of descriptions, as ground truth neuron concepts do not exist for hidden layer neurons. In section 4.1 we show that our method outperforms existing neuron description methods in a large scale crowdsourced study. Next in section 4.2 we explore the different scoring function choices and show why we chose to use *TopK Squared + Image Product*. In section 4.3 we study the importance of our concept selection process, showcasing our method produces very good descriptions even without Best Concept Selection, but concept selection further refines the descriptions. Finally, we provide qualitative examples of our descriptions in Figure 1 and Appendix B.11

## 4.1 CROWDSOURCED EXPERIMENT

**Setup.** Our experiment compares the quality of labels produced by Describe-and-Dissect against 3 baselines: CLIP-Dissect, MILAN, and Network Dissection. For MILAN we used the most powerful *base* model in our experiments.

We dissected both a ResNet-50 network pretrained on Imagenet-1K and ResNet-18 trained on Places365, using the union of ImageNet validation dataset and Broden Bau et al. (2017) as our probing dataset. For both models we evaluated 4 of the intermediate layers(end of each residual block), with 200 randomly chosen neurons per layer for ResNet50 and 50 per layer for ResNet-18. Each neurons description was evaluated by 3 different workers. Note we were unable to compare against Network Dissection on RN18 due to time constraints, but as it was the weakest method on RN50 this should have little impact on results.

The full task interface and additional experiment details are available in Appendix A.3. Workers were presented with the top 10 activating images of a neuron followed by four separate descriptions; each description corresponds to a label produced by one of the four methods compared. The descriptions are rated on a 1-5 scale, where a rating of 1 represents that the user "strongly disagrees" with the given description, and a rating of 5 represents that the user "strongly agrees" with the given description. Additionally, we ask workers to select the description that best represents the 10 highly activating images presented.

**Results.** Table 2 and Table 3 shows the results of a large scale human evaluation study conducted on Amazon Mechanical Turk (AMT). Looking at "% time selected as best" as the comparison metric, our results show that **DnD** performs over $2\times$ better than all baseline methods when dissecting

Table 2: **Averaged AMT results across layers in ResNet-50**. We can see our descriptions are consistently rated the highest, rated between *Agree* and *Strongly Agree*. Our method is also chosen as the best description $> 50\%$ of the time, more than twice as often as the best baseline.

| Metric / Method | Network Dissection | MILAN | CLIP-Dissect | Describe-and-Dissect (**Ours**) |
|---|---|---|---|---|
| Mean Rating | $3.14 \pm 0.032$ | $3.21 \pm 0.032$ | $3.67 \pm 0.028$ | $\mathbf{4.15 \pm 0.022}$ |
| % selected as best | 12.67% | 13.30% | 23.17% | **50.86%** |

Table 3: **Averaged AMT results across layers in ResNet-18**. Similar to Table 2, we can see Describe-and-Dissect still outperforms existing methods ResNet-18 trained on Places365. **DnD** was selected the best method of the $3 > 63\%$ of time time, more than 3 times as often as the second best method.

| Metric / Methods | MILAN | CLIP-Dissect | Describe-and-Dissect (**Ours**) |
|---|---|---|---|
| Mean Rating | $3.27 \pm 0.062$ | $3.45 \pm 0.059$ | $\mathbf{4.16 \pm 0.045}$ |
| % selected as best | 17.61% | 19.19% | **63.21%** |

ResNet-50 and over $3\times$ better when dissecting ResNet-18, being selected the best of the three an impressive 63.21% of the time. In terms of mean rating, our method achieves an average label rating over 4.1 for both dissected models, whereas the average rating for the second best method CLIP-dissect is only 3.67 on ResNet-50 and 3.45 on ResNet-18. Our method also significantly outperforms MILAN's *generative* labels, which averaged below 3.3 for both target models. In conclusion we have shown our method significantly outperforms existing methods in crowdsourced evaluation, and does this consistently across different models and dataset. Detailed comparison of performance per layer is available in the Appendix A.4.

## 4.2 SELECTING A SCORING FUNCTION

**Setup.** We again used ResNet-50 with Imagenet + Broden as the probing dataset. 50 neurons each were randomly chosen from 4 layers, with each neuron evaluated twice, again rating the quality of descriptions on a scale 1-5. The participants in this experiment were volunteers with no knowledge of which descriptions were provided by which methods. Because of the similarity between potential concepts generated by **DnD** Concept Generation (step 2), we also add Network Dissection, MILAN, and CLIP-dissect labels into the set of potential concepts that **DnD** Concept Selection can choose from to increase variance between scoring functions.

Table 4: **Comparison of scoring functions**. Total of 276 evaluations were performed on interpretable neurons (neurons deemed uninterpretable by raters were excluded) from the first 4 layers of ResNet-50 on a scale from 1 to 5. We can see that TopK Squared + Image Products increases overall rating by 10.67% compared to Mean scoring.

| Scoring Function / Layer | Layer 1 | Layer 2 | Layer 3 | Layer 4 | All Layers |
|---|---|---|---|---|---|
| Mean | 3.35 | 2.90 | 2.71 | 3.08 | 3.00 |
| TopK Squared | 3.45 | 2.88 | 2.62 | 3.15 | 3.02 |
| TopK Squared + Image Products | **3.70** | **3.14** | **3.13** | **3.38** | **3.32** |
| Improvement | 10.45% | 8.28% | 15.50% | 9.74% | 10.67% |

Table 5: Human evaluation results for **DnD** (w/o Best Concept Selection) versus full Describe-and-Dissect. Full pipeline improves or maintains performance on every layer in ResNet-50.

| Method / Layer | Layer 1 | Layer 2 | Layer 3 | Layer 4 | **All Layers** |
|---|---|---|---|---|---|
| **DnD** (w/o Best Concept Selection) | **3.54** | 3.77 | 4.00 | 4.02 | 3.84 |
| **DnD** (full pipeline) | **3.54** | **4.00** | **4.24** | **4.13** | **3.97** |

(a) Layer 2 Neuron 312

(b) Layer 3 Neuron 927

Figure 4: **Concept Selection (Step 3) supplements Concept Generation (Step 2) accuracy**. We show that concept selection improves Concept Generation by validating candidate concepts.

**Results.** Table 4 shows the performance of different scoring functions described in section 3.4. Using mean as the baseline scoring function, we show that TopK Squared + Image Products outperforms mean for all four layers in ResNet-50. Across all layers, we observe a significant 10.67% increase in accuracy when compared to the baseline, and similar increase over only using TopK Squared.

### 4.3 ABLATION: THE EFFECT OF CONCEPT SELECTION

Using the same setup as section 4.2, we conducted an ablation study to determine the effectiveness of our Best Concept Selection (step 3) on the pipeline. Table 5 shows the effect of Best Concept Selection on the overall accuracy of **DnD**. We can see DnD performance is already high without Best Concept Selection, but Concept Selection further improves the quality of selected labels Layers 2 through Layer 4, while having the same performance on Layer 1. One potential explanation is due to Layer 1 detecting more limited lower level concepts, there is less variance in candidate descriptions identified in Concept Generation (step 2), resulting in similar ratings across the set of candidate concepts $T$. We can see some individual examples of the improvement Concept Selection provides in Figure 4, with the new labels yielding more specific and accurate descriptions of the neuron, for example Layer2 Neuron 312 becomes more specific *colorful festive settings* instead of generic *Visual Elements*.

## 5 CONCLUSIONS

In this paper, we presented Describe-and-Dissect, a novel method for automatically labeling the functionality of deep vision neurons without the need for labeled training data or a provided concept set. We accomplish this by generating captions for the top activating images of a neuron and combining these captions using natural language processing to create a set of complex "candidate concepts". From this set, we generate a new set of synthetic images using a text-to-image model to refine predictions of Describe-and-Dissect. In addition, we propose TopK Squared Mean + Image Products, a scoring function which utilizes activations from the target model and information in latent image embedding spaces, to select the candidate concept that best represents the ideas encapsulated by a single neuron unit in the target model. Finally, we have shown that Describe-and-Dissect significantly outperforms contemporary baseline methods in a crowdsourced evaluation.

**Reproducibility:** We have described our method and experiment in sufficient detail for reproduction in sections 3 and 4 as well as Appendix A.2 and A.3. In addition, all our code will be released to public before publication.

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

# A APPENDIX

In the appendix, we discuss specifics of the methods and experiments presented in the paper and provide additional results. Section A.1 discusses some potential limitations of **DnD**, Section A.2 details the GPT prompt used in Section 3.2, Section A.3 presents the exact setup of the AMT experiment discussed in Section 4.1, Section A.4 contains more detailed results from our experiment in Section 4.1 with a breakdown by layer, and B.11 gives more qualitative results of our explanations on random neurons, providing many individual neuron examples of our method compared with the baselines.

## A.1 LIMITATIONS

One limitation of Describe-and-Dissect is the relatively high computational cost, taking on average about 38.8 seconds per neuron with a Tesla V100 GPU. However, this problem can be likely well-addressed as the current pipeline has not optimized for speed-purpose yet (e.g. leveraging multiple GPU etc). Another potential limitation of our method is that since it first converts the images into text and then generates its labels based on the text, **DnD** likely cannot detect purely visual elements occasionally found in lower layers of networks, such as contrast or spatial elements. Additionally, our method only takes into account the top $k$ most highly activating images when generating labels. Excluding the information found in lower activating images may not allow our method to gain a full picture of a neuron. However, this problem of only focusing on top activations is shared by all existing methods and we are not aware of good solutions to it. Finally, **DnD** is limited by how well the image-to-text, natural language processing, and text-to-image models are able to perform. However, this also means that future innovations in these types of Machine Learning models can increase the performance of our method.

**Polysemantic neurons:** Existing works Olah et al. (2020); Mu & Andreas (2020); Scherlis et al. (2023) have shown that many neurons in common neural networks are polysemantic, i.e. represent several unrelated concepts or no clear concept at all. This is a challenge when attempting to provide simple text descriptions to individual neurons, and is a limitation of our approach, but can be some-what addressed via methods such as adjusting **DnD** to provide multiple descriptions per neuron as we have done in the Appendix B.9. Another way to address this is by providing an interpretability score, which can differentiate good descriptions from poor ones, which we have explored in the Appendix B.10. We also note that due to the generative nature of **DnD**, even its single labels can often encapsulate multiple concepts by using coordinating conjunctions and lengthier descriptions. However, polysemantic neurons still remain a problem to us and other existing methods such as Bau et al. (2017) Hernandez et al. (2022) Oikarinen & Weng (2023). One promising recent direction to alleviate polysemanticity is via sparse autoencoders as explored by Bricken et al. (2023).

**Challenges in comparing different methods:** Providing a fair comparison between descriptions generated by different methods is a very challenging task. The main method we (and previous work) utilized was displaying highly activating images and asking humans whether the description matches these images. This evaluation itself has some flaws, such as only focusing on the top-k activations and ignoring other neuron behavior. In addition, the question of what are the highly activating images and how to display them is a surprisingly complex one and requires multiple judgement calls, with different papers using different methods with little to no justification for their choices. First, which images are most highly activating. When our neuron is a CNN channel, its activation is 2D so to sort them you need to summarize it to a scalar, typically using either max or average pooling. In our experiments we used average, but max gives a different set of results. Second, the choice of probing dataset is important, as it will affect which images get displayed. Finally choosing if/how to highlight the highly activating region to raters will have an effect. We chose not to highlight as we crop the highly activating regions which can provide similar information. If highlighting is used, choices like what activation threshold to use for highlights and how to display non-activating regions will also affect your results. Different choices for all these parameters will give different results, and it is hard to provide a fully fair comparison, so results such as Section 4.1 should be taken with a grain of salt.

### A.2 GPT PROMPT

The Concept Selection process utilizes GPT to identify and coherently summarize similarities between image captions of every image in $I$. Due to the generative nature of **DnD**, engineering an precise prompt is crucial to the method's overall performance. The use of GPT in **DnD** can be split into two primary cases, each with an individual prompt: A.2.1) Summarizing image captions to better bring out the underlying concepts, and A.2.2) Detecting similarities between the image captions. We explain each part separately.

**A.2.1 Summarizing Image Captions**. BLIP's image captions frequently contain extraneous details that detract from the ideas represented. To resolve this problem, captions are fed through GPT and summarized into simpler descriptions to represent the concept. We use the following prompt:

> "Only state your answer without a period and quotation marks. Do not number your answer. State one coherent and concise concept label that simplifies the following description and deletes any unnecessary details:"

**A.2.2 Detecting Similarities Between Image Captions**. To describe the similarity between image captions, we design the following prompt:

> "Only state your answer without a period and quotation marks and do not simply repeat the descriptions. State one coherent and concise concept label that is 1-5 words long and can semantically summarize and represent most, not necessarily all, of the conceptual similarities in the following descriptions:"

In addition to the prompt, we also feed GPT two sets of example descriptions and along with the respective human-identified similarities:

**Example 1:**

**Human-Identified Similarity**: "multicolored textiles"
**Image Captions**:
- "a purple background with a very soft texture"
- "a brown background with a diagonal pattern of lines and lines"
- "a white windmill with a red door and a red door in the middle of the picture"
- "a beige background with a rough texture of linen"
- "a beige background with a rough texture and a very soft texture"

**Example 2:**

**Human-Identified Similarity**: "red-themed scenes"
**Image Captions**:
- "a little girl is sitting in a red tractor with the word sofy on the front"
- "a toy car sits on a red ottoman in a play room"
- "a red dress with silver studs and a silver belt"
- "a red chevrolet camaro is on display at a car show"
- "a red spool of a cable with the word red on it"

### A.3 AMAZON MECHANICAL TURK SETUP

In this section we explain the specifics of our Amazon Mechanical Turk experiment setup in section 4.1. In the interface show in 5, we display the top 10 highly activating images for a neuron of interest and prompt workers to answer the question: "Does description accurately describe most of the above

Figure 5: **Amazon Mechanical Turk experiment user interface**. For each neuron, we present participants with the top 10 highly activating images and prompt them to rate each method's label based by how accurately the description represents the set of images. We also ask participants to select the best label from the four descriptions presented.

10 images?". Each given description is rated on a 1-5 scale with 1 representing "Strongly Disagree", 3 representing "Neither Agree nor Disagree", and 5 representing "Strongly Agree". Evaluators are finally asked to select the label that best describes the set of 10 images.

We used the final question to ensure the validity of participants' assessments by ensuring their choice for best description is consistent with description ratings, If the label selected as best description in the final question was not (one of) the highest rated descriptions by the user, we deemed a response invalid. In our analysis we only kept valid responses, which amounted to 1744/2400 total responses for Resnet-50 and 443/600 responses for ResNet-18, so around 75% of the responses were valid.

We collected responses from workers over 18 years old, based in the United States who had $> 98\%$ approval rating and more than 10,000 approved HITs to maximize the quality of the responses. We paid workers \$0.08 per response, and our experiment was deemed exempt by the relevant IRB board.

One downside of our experimental setup is that it is not clear which highly activating images we should display to the users and how, and these choices may be important to the final results. In our experiments we displayed the images in $\mathcal{D}_{\text{probe}}$ with the highest mean activation as done by CLIP-DissectOikarinen & Weng (2023), but displaying images with highest max activation would likely be better for MILAN Hernandez et al. (2022). Unfortunately there is no simple answer to which images we should display that would be good for all methods.

Table 6: **Results for individual layers of ResNet-50**. We observe that **DnD** is the best method across all layers in ResNet-50. Our concepts achieved a mean rating of over 4.0 for every layer and were 2× as likely to be selected as the best concept when compared to baseline methods.

| | | Method | | | |
|---|---|---|---|---|---|
| **Metric** | **Layer** | **Network Dissection** | **MILAN** | **CLIP-Dissect** | **Describe-and-Dissect (Ours)** |
| Mean Rating | Layer 1 | $3.41 \pm 0.058$ | $3.41 \pm 0.060$ | $3.63 \pm 0.057$ | **$4.16 \pm 0.041$** |
| | Layer 2 | $3.14 \pm 0.067$ | $3.12 \pm 0.064$ | $3.55 \pm 0.057$ | **$4.07 \pm 0.048$** |
| | Layer 3 | $3.04 \pm 0.066$ | $2.96 \pm 0.066$ | $3.66 \pm 0.055$ | **$4.14 \pm 0.042$** |
| | Layer 4 | $2.97 \pm 0.066$ | $3.34 \pm 0.061$ | $3.82 \pm 0.054$ | **$4.21 \pm 0.044$** |
| % time | Layer 1 | 13.18% | 14.32% | 22.50% | **50.00%** |
| selected | Layer 2 | 15.27% | 12.41% | 19.33% | **52.98%** |
| as best | Layer 3 | 11.82% | 12.73% | 25.00% | **50.45%** |
| answer | Layer 4 | 10.56% | 13.71% | 25.62% | **50.11%** |

## A.4 SECTION 4.1 DETAILED RESULTS

Tables 6 and 7 show the Mechanical Turk ratings for different methods broken down by layer. Our method outperforms all baseline methods for every layer by a significant margin in both ResNet-50 and ResNet-18.

Table 7: **Results for individual layers of ResNet-18**. **DnD** performs significantly better across every layer in ResNet-18 when compared to the baseline methods. Additionally, our concepts are selected over 40% more often than any other method.

| | | Method | | |
|---|---|---|---|---|
| **Metric** | **Layer** | **MILAN** | **CLIP-Dissect** | **Describe-and-Dissect (Ours)** |
| Mean Rating | Layer 1 | $3.20 \pm 0.119$ | $3.17 \pm 0.121$ | **$4.07 \pm 0.101$** |
| | Layer 2 | $3.12 \pm 0.133$ | $3.47 \pm 0.122$ | **$4.22 \pm 0.086$** |
| | Layer 3 | $3.00 \pm 0.128$ | $3.43 \pm 0.117$ | **$4.18 \pm 0.081$** |
| | Layer 4 | $3.71 \pm 0.107$ | $3.70 \pm 0.104$ | **$4.18 \pm 0.092$** |
| % time | Layer 1 | 16.51% | 20.18% | **63.30%** |
| selected | Layer 2 | 16.67% | 15.74% | **67.59%** |
| as best | Layer 3 | 15.24% | 20.95% | **63.81%** |
| answer | Layer 4 | 21.49% | 19.83% | **58.68%** |

## A.5 QUALITATIVE RESULTS

Figures 6, 7, 8, and 9 provide supplementary qualitative results for **DnD** along with results produced by baseline methods for ResNet-50 and ResNet-18.

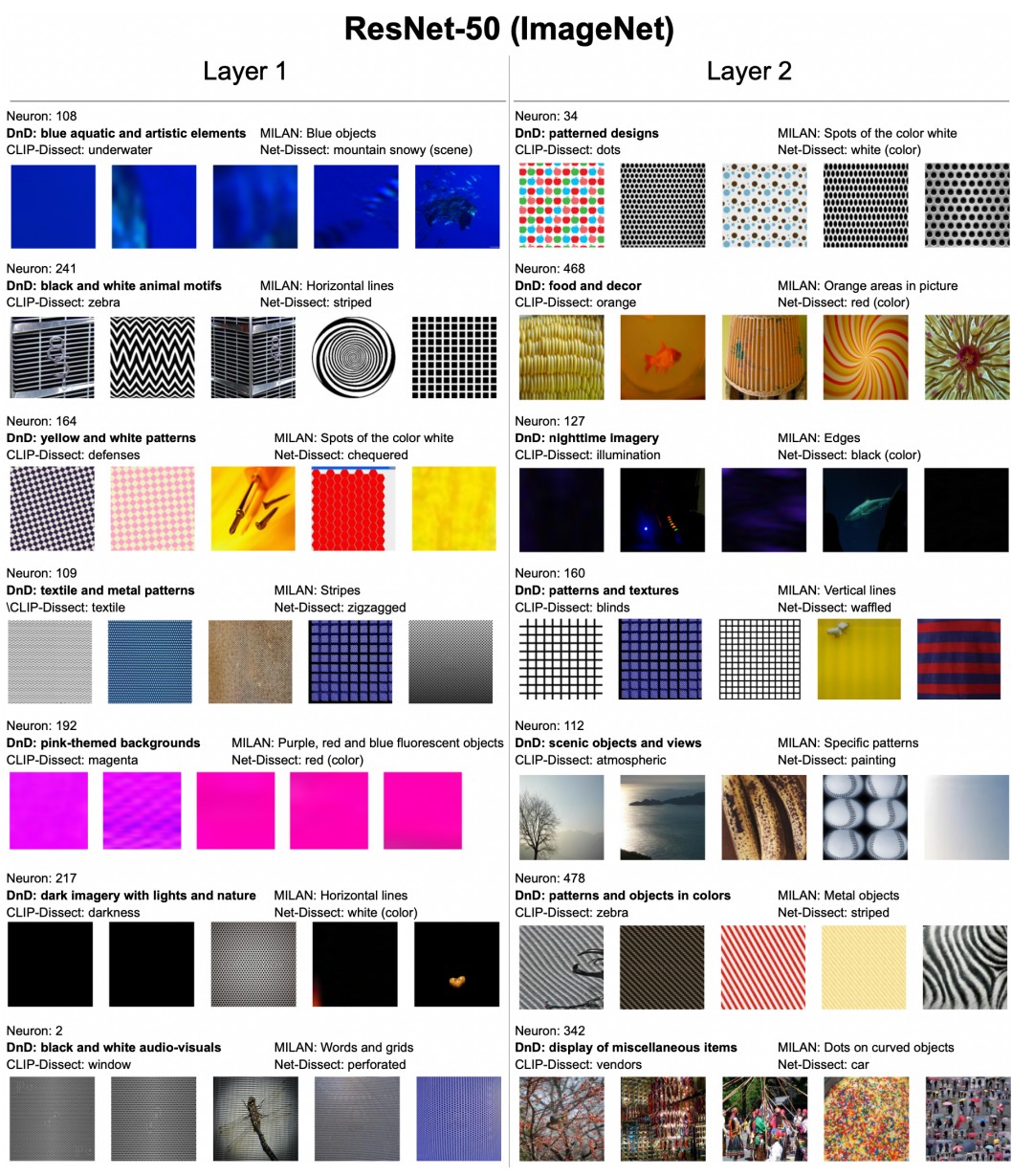

Figure 6: **Additional examples of DnD results from Layer 1 and 2 of ResNet-50**. We showcase a set of randomly selected neurons and their descriptions from Layer 1 and 2 of ResNet-50 trained on ImageNet-1K.

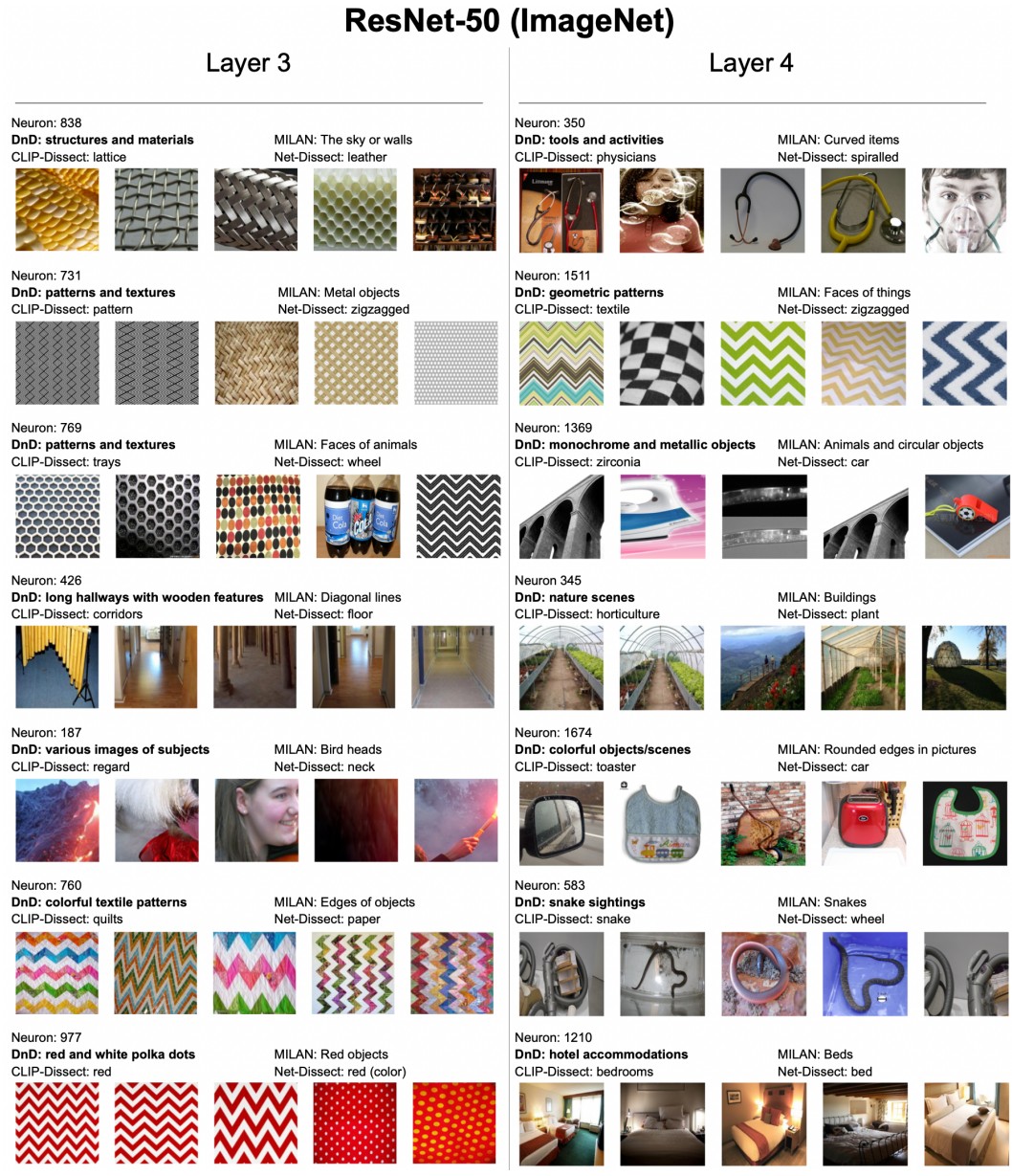

Figure 7: **Additional examples of DnD results from Layer 3 and 4 of ResNet-50**. We showcase a set of randomly selected neurons and their descriptions from Layer 3 and 4 of ResNet-50 trained on ImageNet-1K.

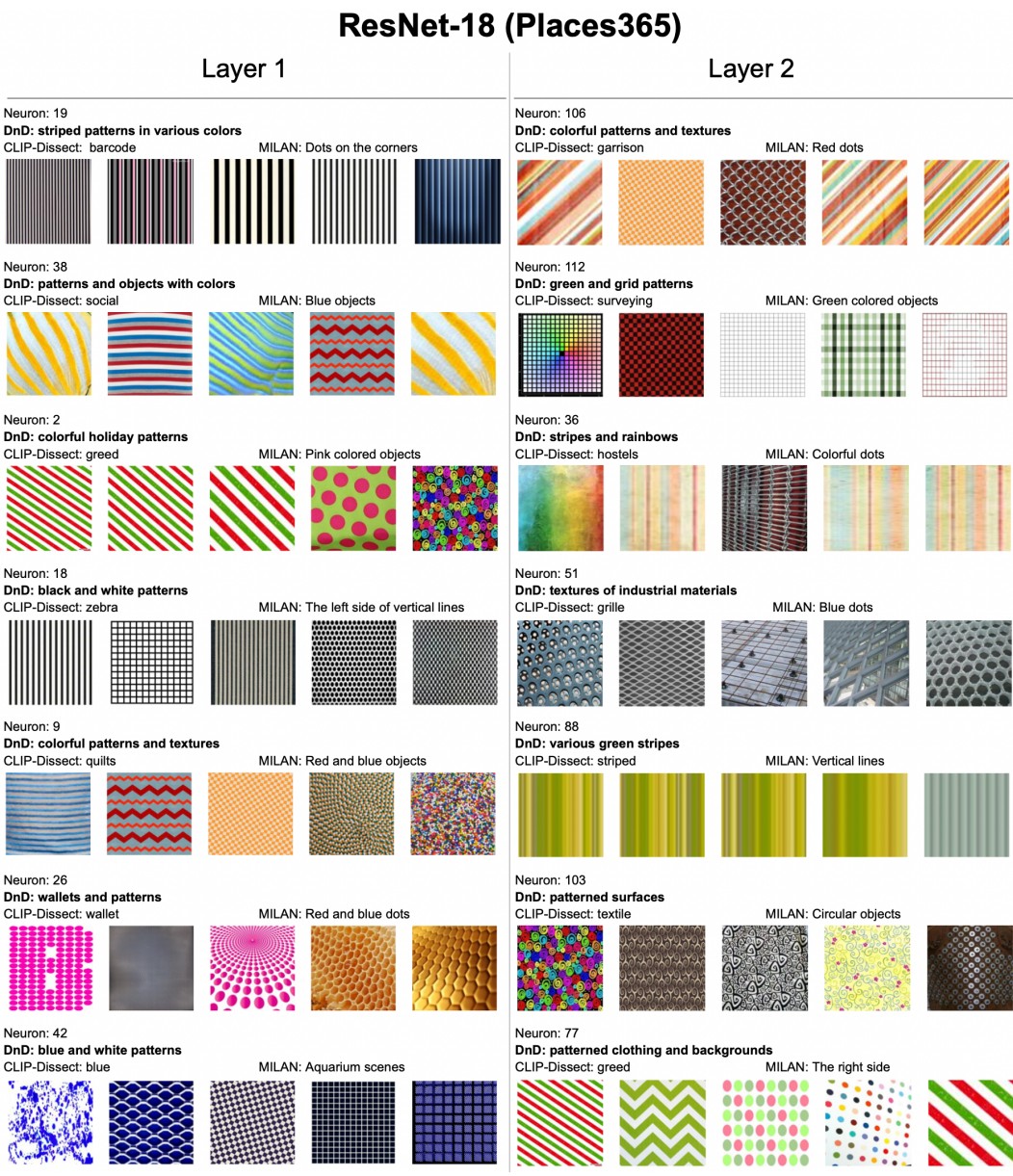

Figure 8: **Additional examples of DnD results from Layer 1 and 2 of ResNet-18**. We showcase a set of randomly selected neurons and their descriptions from Layer 1 and 2 of ResNet-18 trained on Places365.

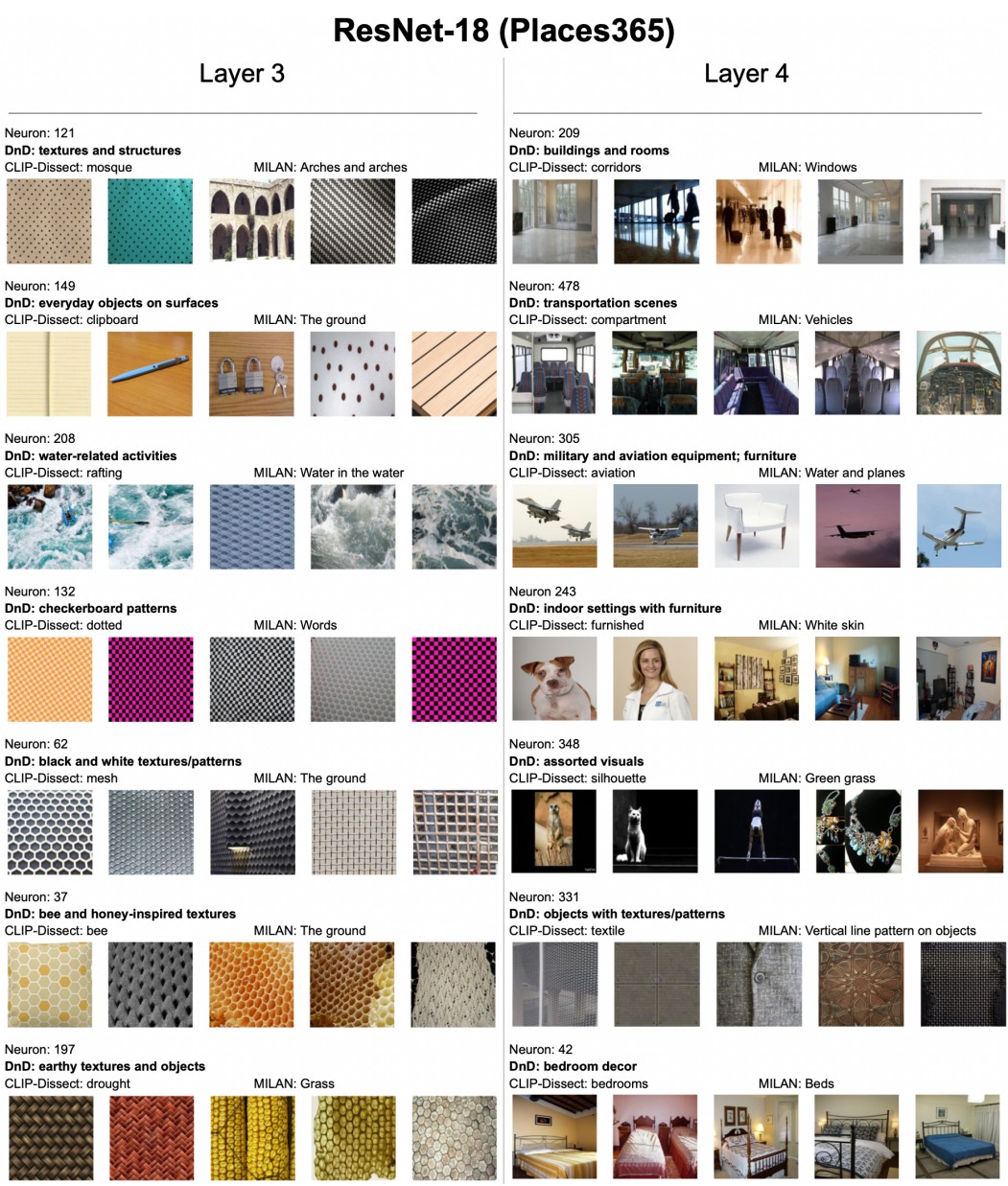

Figure 9: **Additional examples of DnD results from Layer 3 and 4 of ResNet-18**. We showcase a set of randomly selected neurons and their descriptions from Layer 3 and 4 of ResNet-18 trained on Places365.

# B  REBUTTAL EXPERIMENTS

## B.1  DETAILED DESCRIPTION OF ATTENTION CROPPING

The attention cropping algorithm described in Sec 3.1 constitutes of three primary procedures:

- **Step 1.** Compute the optimal global threshold, $\lambda$ for the activation map of a given neuron $n$ using Otsu's Method Otsu (1979).
- **Step 2.** Highlight contours of salient regions on the activation map that activate higher than $\lambda$.
- **Step 3.** Crop $\alpha$ largest salient regions from the original activating image that have an IoU score less than $\eta$ with all prior cropped regions.

Here, we provide more implementation details for each step:

- **Step 1. Global Thresholding using Otsu's Method.** For an activation map of neuron $n$, we define regions with high activations as the "foreground" and regions with low activations as the "background". Otsu's Method Otsu (1979) automatically calculates threshold values which maximizes interclass variance between foreground and background pixels. Interclass variance $\sigma_B^2$ is defined by $\sigma_B^2 = W_b W_f (\mu_b - \mu_f)^2$, where $W_{b,f}$ denotes weights of background and foreground pixels and $\mu_{b,f}$ denotes the mean intensity of background and foreground pixels respectively. For global thresholding used in **DnD**, $W_f$ is the fraction of pixels in activation map $M_n(x_i)$ above potential threshold $\lambda$ while $W_b$ is the fraction of pixels below $\lambda$. A value of $\lambda$ that maximizes $\sigma_B^2$ is used as the global threshold.
- **Step 2. Highlight Contours of Salient Regions.** **DnD** utilizes OpenCV's contour detection algorithm to identify changes within image color in the binary masked activation map. Traced contours segments are compressed into four end points, where each end point represents a vertex of the bounding box for the salient region.
- **Step 3. Crop Activating Images.** Bounding boxes recorded from **Step 2.** are overlaid on corresponding activating images from $D_{probe}$, which are cropped to form $D_{cropped}$. To prevent overlapping attention crops, the IoU score between every crop in $D_{cropped}$ is less than an empirically set parameter $\eta$, where high $\eta$ values yield attention crops with greater similarities.

We also visualize these steps in the below Figure 10 for better understanding these steps.

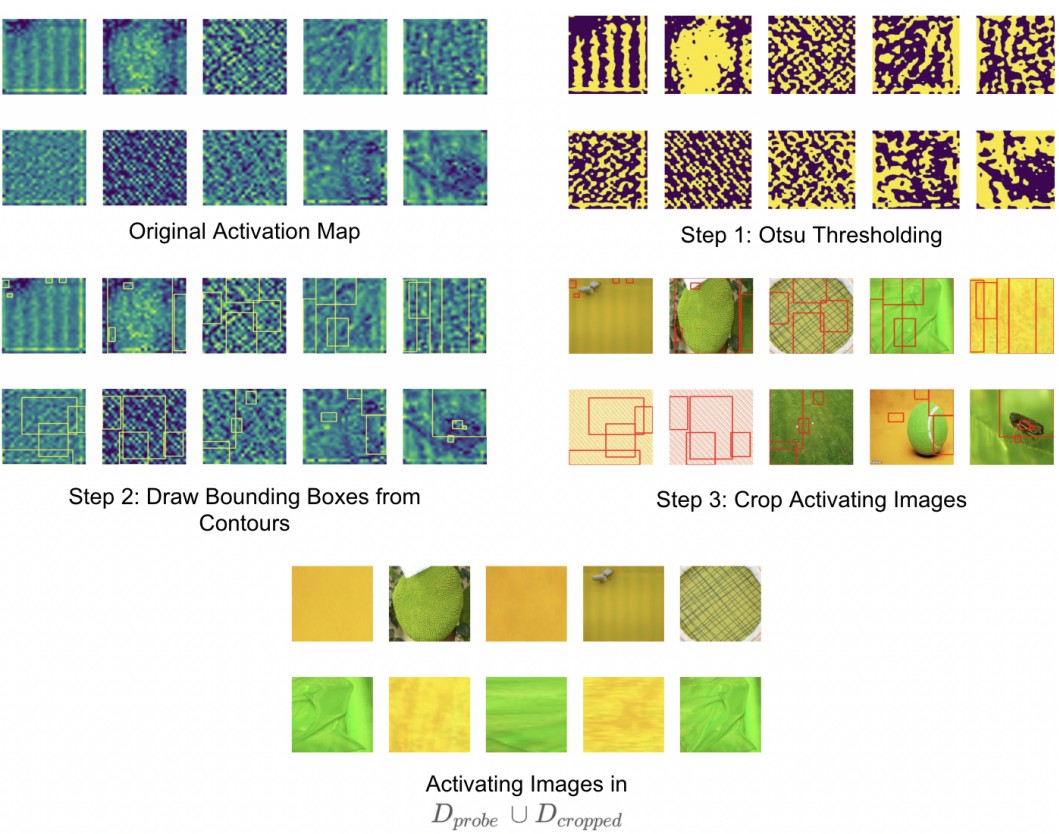

Figure 10: **Detailed Visualization of Attention Cropping Pipeline.** All three steps of attention cropping are shown for Layer 2 Neuron 165. **Steps 1 and 2** illustrate the derivation of bounding boxes from salient regions in the original activation map and are overlaid on original activating images from $D_{probe}$ in **Step 3**.

## B.2 ABLATION: ATTENTION CROPPING

Attention cropping is a critical component of **DnD** due to generative image-to-text caption. Unlike models that utilize fixed concept sets such as CLIP-dissect Oikarinen & Weng (2023), image captioning models are prone to spuriously correlated concepts which are largely irrelevant to a neuron's activation. To determine the effects of attention cropping on subsequent processes in the **DnD** pipeline, we evaluate **DnD** on $D_{probe}$ without augmented image crops from $D_{cropped}$. We show qualitative examples of this effect in Figure 11.

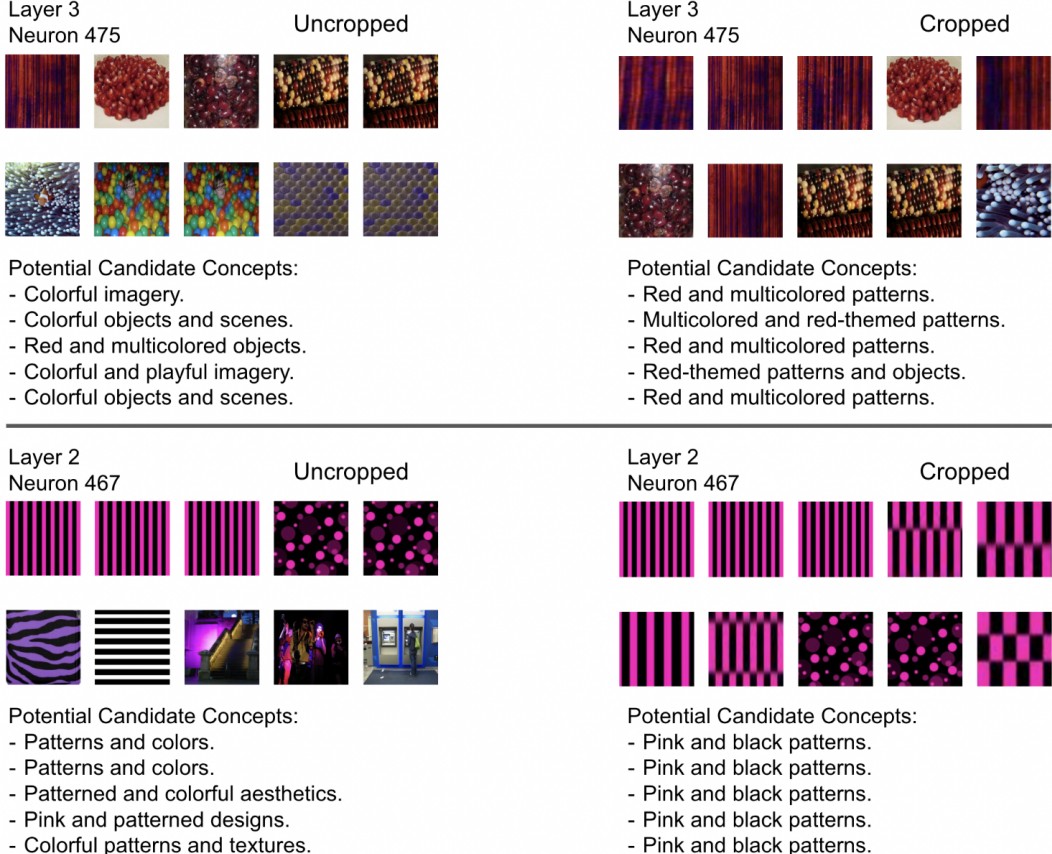

Figure 11: **Examples where Attention Cropping Improves DnD.** The left panel shows the result without attention cropping (**Uncropped**) while the right panel show the result with attention cropping (**Cropped**). It can be seen that Attention cropping eliminates spurious features in highly activating images and improves accuracy of potential candidate concepts.

## B.3  ABLATION: IMAGE-TO-TEXT MODEL

In light of advancements in Image-to-Text models, we compare BLIP Li et al. (2022) to a more recently released model, BLIP-2 Li et al. (2023). Unlike BLIP, BLIP-2 leverages frozen image encoder and LLMs while introducing Querying Transformer to bridge the modality gap between two models. We experiment with using BLIP-2 as the Image-to-Text model and quantitatively compare with BLIP by computing the mean cosine similarity between the best concept chosen from Concept Selection. As discussed in the Appendix B.7, CLIP-ViT-B/16 cosine similarity is a stronger indicator of conceptual connections than all-mpnet-base-v2 similarity for generative labels. Accordingly, CLIP-ViT-B/16 cosine similarity is used as the comparison metric. We evaluate on 50 randomly chosen neurons from each intermediate layer of RN50 and results of the experiment are detailed in Table 8. From Table 8, BLIP and BLIP-2 produce highly correlated concepts across all four layers of RN50 and a 84.8% similarity across the entire network.

Table 8: **Mean Cosine Similarity Between BLIP and BLIP-2 Labels.** For each layer in RN50, we compute the mean CLIP cosine similarity between BLIP and BLIP-2 labels for 50 randomly chosen neurons. Similar conceptual ideas between both models are reflected in the high similarity scores.

| Metric | Layer 1 | Layer 2 | Layer 3 | Layer 4 | All Layers |
|---|---|---|---|---|---|
| CLIP cos | 0.839 | 0.844 | 0.842 | 0.868 | 0.848 |

We also show examples of similar description produced in Figure 12. Somewhat counter-intuitively, our qualitative analysis of neurons in preliminary layers of RN50 reveals examples where BLIP-2 fails to detect low level concepts that BLIP is able to capture. Such failure cases limit **DnD** accuracy by generating inaccurate image captions and adversely affects the ability of GPT summarization to identify conceptual similarities between captions. We show two such examples in below Figure 13. In general this experiment shows that our pipeline is not very sensitive to the choice of image-captioning model.

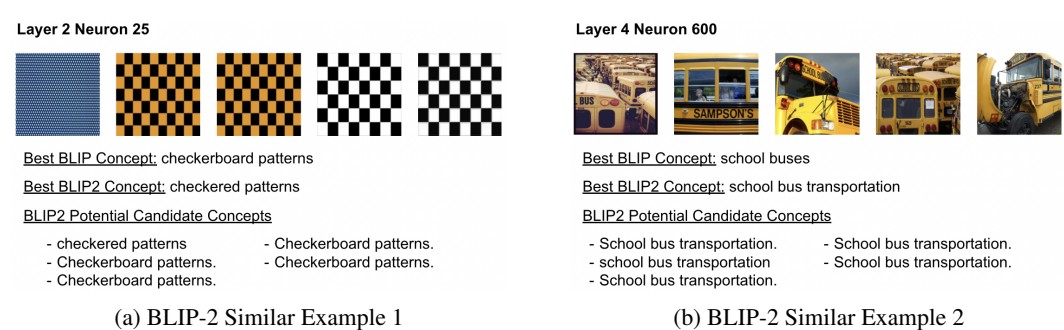

(a) BLIP-2 Similar Example 1             (b) BLIP-2 Similar Example 2

Figure 12: **Examples of Similar BLIP and BLIP-2 Labels.** BLIP and BLIP-2 detect similar concepts across all layers of RN50. Two such examples are detailed above. The CLIP cosine similarity between the Best BLIP Concept and Best BLIP2 Concept for Figure 12a is 0.9722. The similarity for Figure 12b is 0.9077.

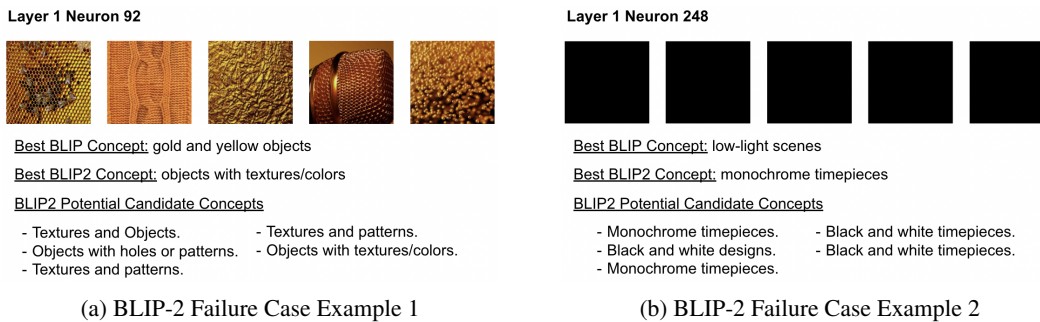

(a) BLIP-2 Failure Case Example 1       (b) BLIP-2 Failure Case Example 2

Figure 13: **Examples of BLIP-2 failure cases.** BLIP-2 overlooks crucial low level concepts in early layers of RN50. Potential Candidate Concepts generated are spuriously correlated and yield poor results in the final **DnD** label. The CLIP cosine similarity between the Best BLIP Concept and Best BLIP2 Concept for Figure 13a is 0.7988. The similarity for Figure 13b is 0.6987.

### B.4 ABLATION: IMAGE CAPTIONING WITH FIXED CONCEPT SET

To analyze the importance of using generative Image-to-Text model, we explore instead utilizing fixed concept sets with CLIP Radford et al. (2021) to generate descriptions for each image instead of BLIP, while the rest of the pipeline is kept the same, i.e. using GPT to summarize etc. For the experiment, we use CLIP-ViT-B/16, where we define $L(\cdot)$ and $E(\cdot)$ as text and image encoders respectively. From the initial concept set $\mathcal{S} = \{t_1, t_2, ...\}$, the best concept for image $I_m$ is defined as $t_l$, where $l = \arg\max_i (L(t_i) \cdot E(I_m)^\top)$. Following CLIP-dissect Oikarinen & Weng (2023), we use $\mathcal{S} = 20\text{k}^2 (20,000$ most common English words) and $D_{probe} = \text{ImageNet} \cup \text{Broden}$.

To compare the performance, following Oikarinen & Weng (2023) we use our model to describe the final layer neurons of ResNet-50 (where we know their ground truth role) and compare descriptions similarity to the class name that neuron is detecting, and as discussed in Appendix B.6. Results in Table 9 show that both methods perform similarly on the FC layer. In intermediate layers, we notice that single word concept captions from 20k significantly limit the expressiveness of **DnD**, and that having generative image descriptions is important for our overall performance. Qualitative examples are shown in Figure 14 and Figure 15 showcases some failures of using CLIP descriptions.

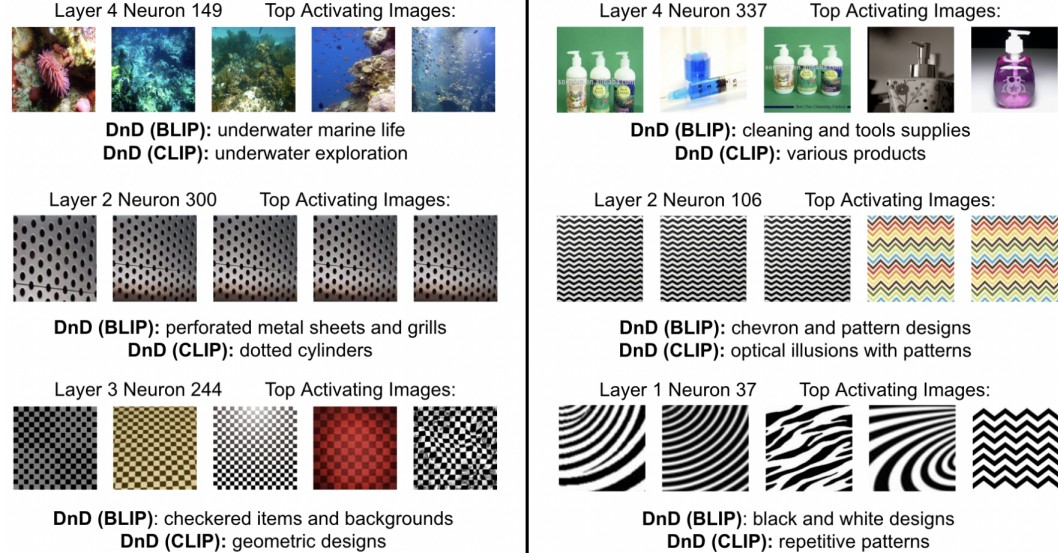

Figure 14: **Examples of DnD with CLIP Image Captioning Compared to DnD (with BLIP).** Despite producing similar results on the FC layer, we see that **DnD** (with BLIP) outperforms **DnD** with CLIP image captioning, especially on intermediate layer neurons. Single word captions fail to fully encapsulate concepts expressed in these layers, resulting in poor **DnD** performance.

Table 9: **Mean FC Layer Similarity of CLIP Captioning.** Utilizing a fixed concept set ($t = 20\text{k}$) to caption activating images via CLIP Radford et al. (2021), we compute the mean cosine similarity across fully connected layers of RN50. Final labels that do not express concepts (see Figure 15) are given a similarity score of 0. With both CLIP cosine similarity and MPNet cosine similarity, we find the performance of **DnD** w/ CLIP Captioning is slightly worse than BLIP generative caption.

| Metric / Methods | Describe-and-Dissect (**Ours**) | DnD w/ CLIP Captioning | % Decline |
|---|---|---|---|
| CLIP cos | **0.7598** | 0.7583 | 0.197% |
| mpnet cos | **0.4588** | 0.4465 | 2.681% |

---

[2]Source: https://github.com/first20hours/google-10000-english/blob/master/20k.txt

**Layer FC Neuron 94**

**DnD w/ CLIP Output:**

Repetition of "hummer" does not provide enough information to determine a coherent and concise concept label. please provide more context or descriptions

**DnD w/ BLIP Output:**       **Ground Truth:**

hummingbird sightings          hummingbird

(a) CLIP Image Captioning Failure Case 1

**Layer FC Neuron 173**

**DnD w/ CLIP Output:**

as an ai language model, i cannot provide any additional information or change the set of descriptions. please provide more context or information for me to generate a coherent and concise concept label

**DnD w/ BLIP Output:**       **Ground Truth:**

dogs and their activities       Ibizan hound, Ibizan Podenco

(b) CLIP Image Captioning Failure Case 2

Figure 15: **Failure Cases of CLIP Image Captioning.** Due to the lack of expressiveness of static concept sets, GPT summarization fails to identify conceptual similarities within CLIP image captions. With dynamic concept sets generated by BLIP, the issue is largely avoided.

## B.5 ABLATION: EFFECTS OF GPT CONCEPT SUMMARIZATION

**DnD** utilizes OpenAI's GPT-3.5 Turbo to summarize similarities between the image caption generated for each $K$ highly activating images of neuron $n$. This process is crucial component to improve accuracy and understandability of our explanations, as shown in the ablation studies below.

As mentioned in Appendix A.2, GPT summarization is composed of two functionalities: **1.** simplifying image captions and **2.** detecting similarities between captions. Ablating away GPT, we substitute the summarized candidate concept set $T$ with the image captions directly generated by BLIP for the $K$ activating images of neuron $n$. We note two primary drawbacks:

- **Poor Concept Quality.** An important aspect of GPT summarization is the ability to abbreviate nuanced captions into semantically coherent concepts (function **1.**). In particular, a failure case for both BLIP and BLIP-2, shown in Figure 16a, is the repetition of nonsensical elements within image captions. Processing these captions are computationally inefficient and can substantially extend runtime. **DnD** resolves this problem by preprocessing image captions prior to similarity detection.

- **Concept specific to a single image.** Without the summarization step, concepts only describe a single image, and fail to capture the shared concept across images. We can see for example that the neuron in Figure 16b gets only described as a "pile of oranges", missing the overall theme of orange color.

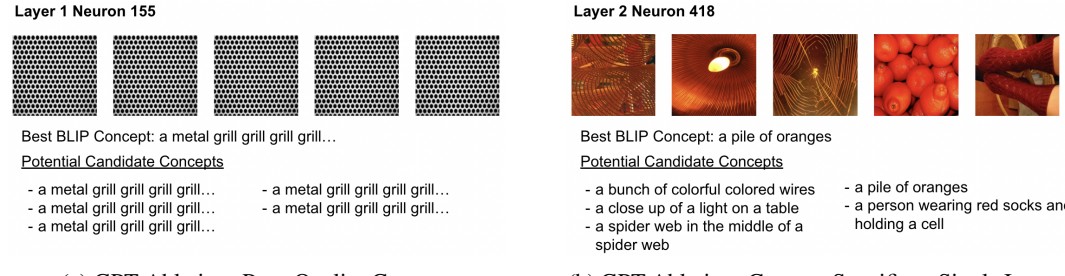

|  (a) GPT Ablation: Poor Quality Concept | (b) GPT Ablation: Concept Specific to Single Image |

Figure 16: **Failure Cases in GPT-Ablation Study.** Figure 16a illustrates potential failure cases of **DnD** without GPT summarization. BLIP produces illogical image captions that hinder overall interpretability. Figure 16b shows an additional failure case where the omission of concept summarization causes individual image captions to generalize to the entire neuron.

## B.6 QUANTITATIVE RESULTS: FINAL LAYER

Though our results are primarily on **DnD**'s results for hidden layers of a network, our method can also be utilized for final layer neurons. Note for final layer neurons we can't augment the probing set with activation crops as the neurons have scalar activations. Table 10 displays the results of our experiment on all of the neurons of ResNet-50's final fully-connected layer, following Oikarinen & Weng (2023) to compare against ground truth neuron role. We compare against MILAN, as it is the other generative contemporary work. We don't include other works with concept sets as the "ground truth" can be included in their concept set or very similar to elements of it, giving them an advantage. Our results show that **DnD** outperforms MILAN, having a greater average CLIP cosine similarity by 0.0518 and a greater average mpnet cosine similarity by 0.18. On average **DnD**'s labels are significantly closer to the ground truths than MILAN's.

Table 10: **Cosine similarity between predicted labels and ground truths on the fully-connected layer of ResNet-50 trained on ImageNet**. We can see Describe-and-Dissect outperforms MILAN.

| Metric / Methods | MILAN | Describe-and-Dissect (**Ours**) |
|---|---|---|
| CLIP cos | 0.7080 | **0.7598** |
| mpnet cos | 0.2788 | **0.4588** |

## B.7 QUANTITATIVE RESULTS: MILANNOTATIONS

In this section we discuss using the MILANNOTATIONS dataset Hernandez et al. (2022) as a set of ground truths to calculate a quantitative results on the intermediate layers of ResNet-152. Table 11 displays an experiment done on 103 randomly chosen "reliable" neurons across the 4 intermediate layers of ResNet-152. A neuron was deemed "reliable" if out of its three corresponding MILANNO-TATIONS, two had a CLIP cosine similarity exceeding a certain threshold (for the purposes of this experiment we set that threshold to 0.81). For this experiment, we define our summary function $g$ to be spatial max, as this is what was used to calculate the highest activating images for the MILANNO-TATIONS. We compare against MILAN trained on places365 and CLIP-Dissect, using the spatial max summary function for CLIP-Dissect as well. Our results are generally mixed, with MILAN performing the best (average of 0.7698), then DnD (average of 0.7543), and finally CLIP-Dissect (average of 0.7390) using CLIP similarity. But with MPNet similarity, CLIP-Dissect is calculated as the best (average of 0.1970) followed by MILAN (average of 0.1391) and then DnD (average of 0.1029). However, the MILANNOTATIONS dataset is very noisy as seen in Figure 17 and thus is largely unreliable for evaluation. The dataset provides three annotations per neuron, each made by a different person. This causes them to often be very different from one another, not representing the correct concept of the neuron. We can see in the table that the average CLIP cosine similarity between the annotations for each of these reliable neurons is 0.7884 and the average MPNet co-sine similarity is 0.1215. Due to this noisyness, we found that the best performing description is as generic as possible as it will be closest to all descriptions. In fact, we found that a trivial baseline describing each neuron as "image" irrespective of the neuron's function outperforms all description methods, including MILAN. This leads us to believe MILANNOTATIONS should not be relied on to evaluate description quality. Figure 17 showcases descriptions and MILANNOTATIONS for some neurons.

Table 11: **Cosine similarity between predicted labels and "ground truth" MILANNOTATIONS on the 4 intermediate layers of ResNet-152 trained on ImageNet**. We observe that when using CLIP similarity, MILAN performs the best followed by **DnD** and then CLIP-Dissect. Simple label-ing every neuron as "Images" outperforms all other methods, demonstrating unreliable of MILAN-NOTATIONS as an evaluation method. The set's noisiness is also shown by the average similarities between the different annotations for each neuron, showing that the annotations are not consistent.

| Metric | Layer | Method | | | | Avg Cos Sim Between Annotations |
| | | MILAN | CLIP-Dissect | Describe-and-Dissect (Ours) | "Images" | |
| --- | --- | --- | --- | --- | --- | --- |
| CLIP cos | Layer 1 | 0.7441 | 0.7398 | 0.7306 | **0.7787** | 0.7643 |
| | Layer 2 | 0.7627 | 0.7472 | 0.7535 | **0.8084** | 0.7976 |
| | Layer 3 | 0.7893 | 0.7414 | 0.7645 | **0.7915** | 0.8026 |
| | Layer 4 | 0.7739 | 0.7291 | 0.7622 | **0.7917** | 0.7845 |
| | All Layers | 0.7698 | 0.7390 | 0.7543 | **0.7924** | 0.7884 |
| mpnet cos | Layer 1 | 0.0801 | 0.1962 | 0.1076 | **0.2374** | 0.1124 |
| | Layer 2 | 0.1143 | 0.1850 | 0.0952 | **0.2382** | 0.1167 |
| | Layer 3 | 0.1462 | 0.2039 | 0.1118 | **0.2356** | 0.1279 |
| | Layer 4 | 0.1973 | 0.1995 | 0.0955 | **0.2382** | 0.1252 |
| | All Layers | 0.1391 | 0.1970 | 0.1029 | **0.2372** | 0.1215 |

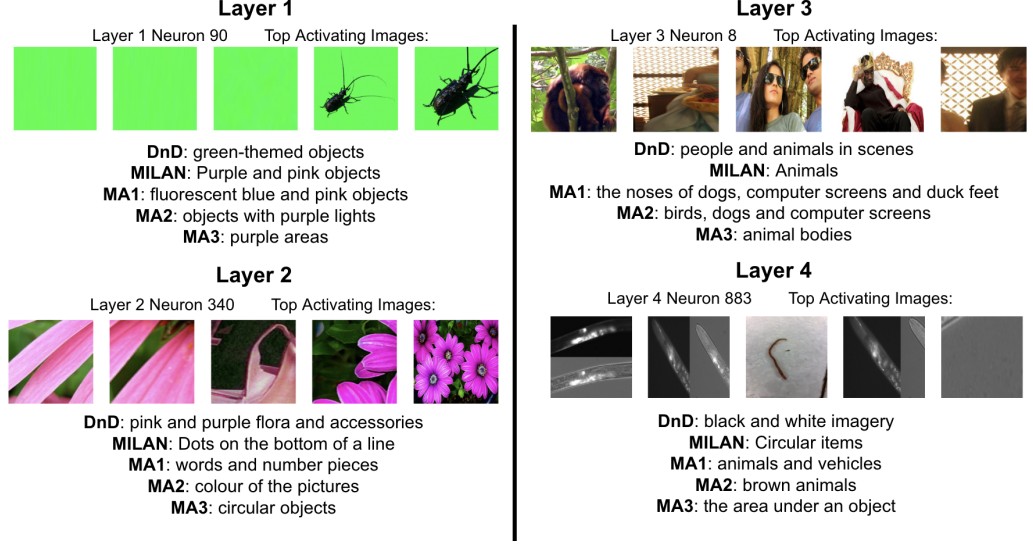

Figure 17: **MILANNOTATIONS from random "interpretable" neurons in ResNet-152 (ImageNet)** as defined in Section: B.7. We showcase a set of randomly selected neurons alongside their corresponding **DnD** label, MILAN label, and its 3 MILANNOTATIONS labels (MA). We can see that the three MILANNOTATIONS for each neuron often don't match up too well and don't accurately describe the neuron.

## B.8 APPLICATION: FINDING CLASSIFIERS FOR NEW CONCEPTS

To showcase a potential use case for neuron descriptions (and provide another way to quantitatively compare explanation methods), we experimented with using neuron descriptions to find a good classifier for a class missing from the training set. Our setup was as follows: we explained all neurons in layer4 of ResNet-50(ImageNet) using different methods. We then wanted to find neurons in this layer that could serve as the best classifiers for an unseen class, specifically the classes in CIFAR-10 and CIFAR-100 datasets. Note there is some overlap between these and ImageNet classes, but CIFAR classes are typically much more broad. To find a neuron to serve as a classifier, we found the neuron whose description was closest to the CIFAR class name in a text embedding space (ensemble of CLIP ViT-B/16 text encoder and mpnet text encoders used in this paper). We then measured how well that neuron(its average activation) performs as a single class classifier on the CIFAR validation dataset, measured by area under ROC curve. For cases where multiple neurons shared the closest description, we averaged the performance of all neurons with that description. Results are shown in Table 12. We can see Describe-and-Dissect performs quite well, reaching AUROC values around 0.75, while MILAN performs much worse. We believe this may be because MILAN descriptions are very generic(likely caused by noisy dataset, see Appendix B.7), which makes it hard to find a classifier for a specific class. We think this is a good measure of explanation quality, as different methods are dissecting the same network, and even if no neurons exist that can directly detect a class, a better method should find a closer approximation.

|          | MILAN  | Describe-and-Dissect(Ours) |
|----------|--------|----------------------------|
| CIFAR10  | 0.5906 | **0.7375**                 |
| CIFAR100 | 0.6514 | **0.7607**                 |

Table 12: The average classification AUC on out of distribution dataset when using neurons with similar description as a classifier. We can see Describe-and-Dissect clearly outperforms MILAN, the only other generative description method.

### B.9 MULTIPLE LABELS

As discussed in the Appendix A.1, many neurons in intermediate layers can be considered "polyse-mantic", meaning that they encode multiple unrelated concepts. Our primary pipeline only produces one label, and though this label can encapsulate more than one concept, providing multiple labels can better account for issues of polysemanticity. We accomplish this by taking the top 3 candidate concepts selected by Best Concept Selection as our final labels. If two of these three final labels have a CLIP cosine similarity exceeding 0.81, we only take the top one of that pair and eliminate the other from the set of final labels. This allows use to bring out accurate labels that account for distinct concepts prevalent in the neuron. We can see this in Figure 18. For neuron 508 from layer 2, **DnD** captures not only the polka dot and metal texture concepts of the neuron, but adds that the textures are primarily black and white. For neuron 511 from layer 3, **DnD** labels the neuron as detecting not only interior elements, but also specifying that these elements are mostly black and white.

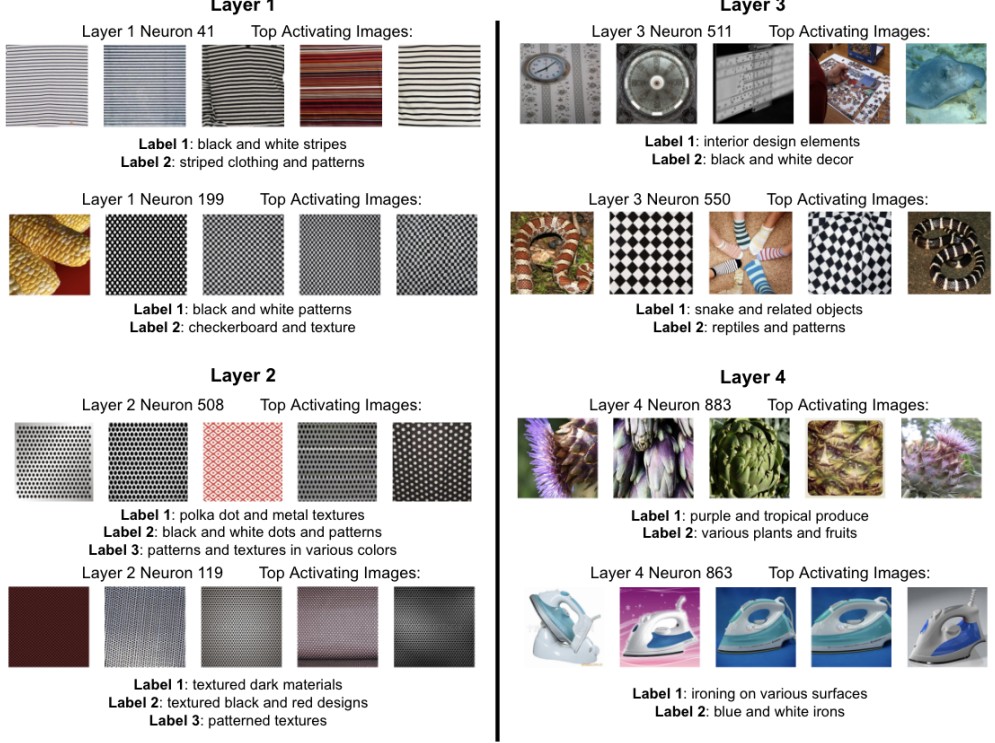

Figure 18: **Example results on ResNet-50 (ImageNet) from the DnD pipeline being modified to produce multiple labels**. We showcase a set of randomly selected neurons to exemplify **DnD**'s capability to provide multiple descriptions.

### B.10 QUANTIFYING NEURON INTERPRETABILITY

As discussed in limitations in Appendix A.1, certain neurons within computer vision networks are "uninterpretable", i.e. their concepts are unintelligible to humans.

As such, it is important that a neuron description method can tell which neurons are interpretable to begin with, so we know whether we should rely on the description. In this section, we explore a quantitative method to compute the interpretability of a neuron by calculating the mean similarity between the set of top activating images $I$. This relies on the assumption that an interpretable, monosemantic neuron will have highly activating images that are semantically similar.

Let $Q = \{E(x_i), x_i \in I, |I| = K\}$ represent the set of embeddings for the top $K$ activating images of neuron $n$, where we use CLIP-ViT-B/16 Radford et al. (2021) as the image encoder $E(\cdot)$. The interpretability of a neuron, denoted $\mu_n$ is defined as the average dot product between the embeddings of its highly activating images:

$$\mu_n = \sum_{q_i \in Q} \sum_{q_j \in Q} \frac{q_i \cdot q_j}{K^2}$$

Larger values of $\mu_n$ represent a greater interpretability of the neuron. Figure 19a shows examples of uninterpretable neurons and Figure 19b shows examples of interpretable neurons. From qualitative evaluations, neurons with $0.85 < \mu_n$ are typically interpretable.

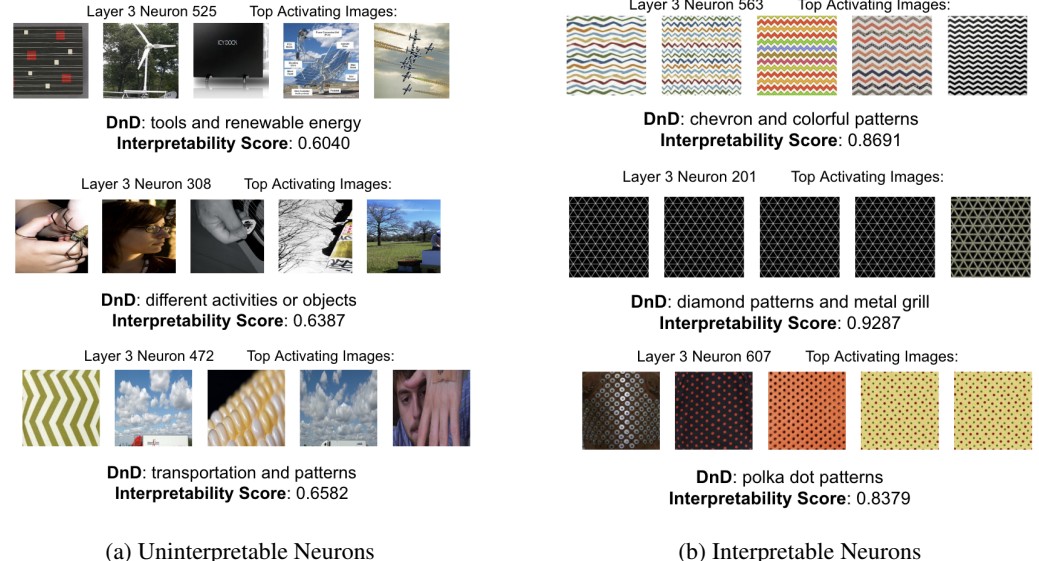

(a) Uninterpretable Neurons                (b) Interpretable Neurons

Figure 19: **Interpretability scores on interpretable and uninterpretable neurons.** We showcase two random uninterpretable neurons and two random interpretable neurons from layer 3 of ResNet-50 (ImageNet). In Figure 19a, we can see that these neurons cannot be assigned clear comprehensive labels and our lower interpretability scores reflect that. In Figure 19b, we can see that these neurons represent clear concepts and our higher interpretability scores reflect that.

### B.11 ADDITIONAL QUALITATIVE RESULTS

Figure 20 showcases our descriptions on random neurons of the ViT-B/16 Vision Transformer trained on ImageNet, showing good descriptions overall, and higher quality than CLIP-Dissect descriptions. Finally in Figures 21 and 22 we dissect ResNet-50 using a different probing dataset, showcasing our method still perfoms well with a different probing dataset.

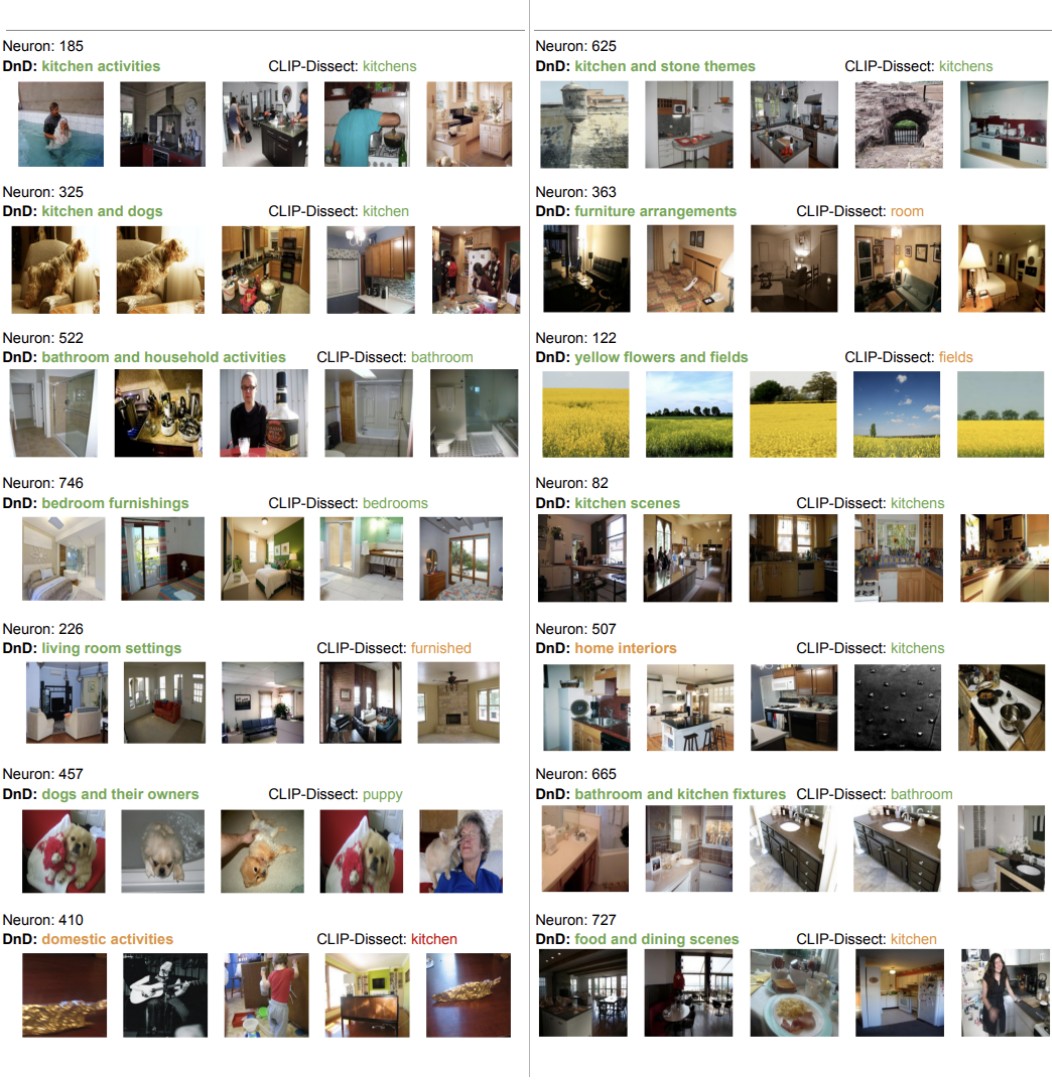

Figure 20: **Examples of DnD results from the encoder layer of ViT-B/16 (ImageNet)**. We showcase a set of randomly selected neurons and their descriptions from the encoder layer of ViT-B/16 trained on ImageNet. **DnD** is model agnostic and can be generalized to any convolutional network. Labels are color-coded by whether we believed they were accurate, somewhat correct/vague or imprecise.

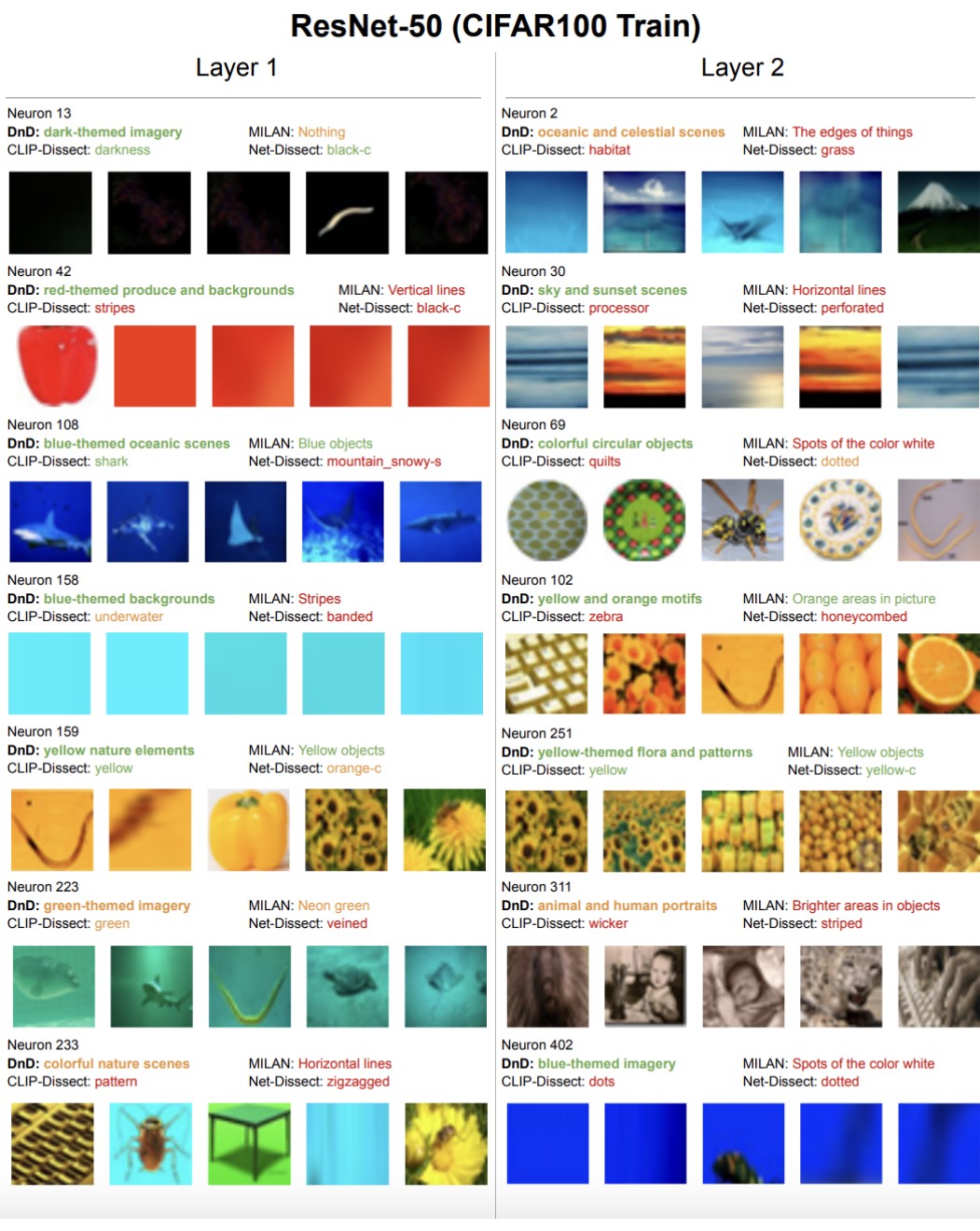

Figure 21: **Examples of DnD results from Layer 1 and 2 of ResNet-50 using the training dataset of CIFAR100 as the probing image set.** We showcase a set of randomly selected neurons and their descriptions from Layer 1 and 2 of ResNet-50 trained on ImageNet-1K. **DnD** is probing dataset agnostic and can maintain its high performance on other image sets. Labels are color-coded by whether we believed they were accurate, somewhat correct/vague or imprecise.

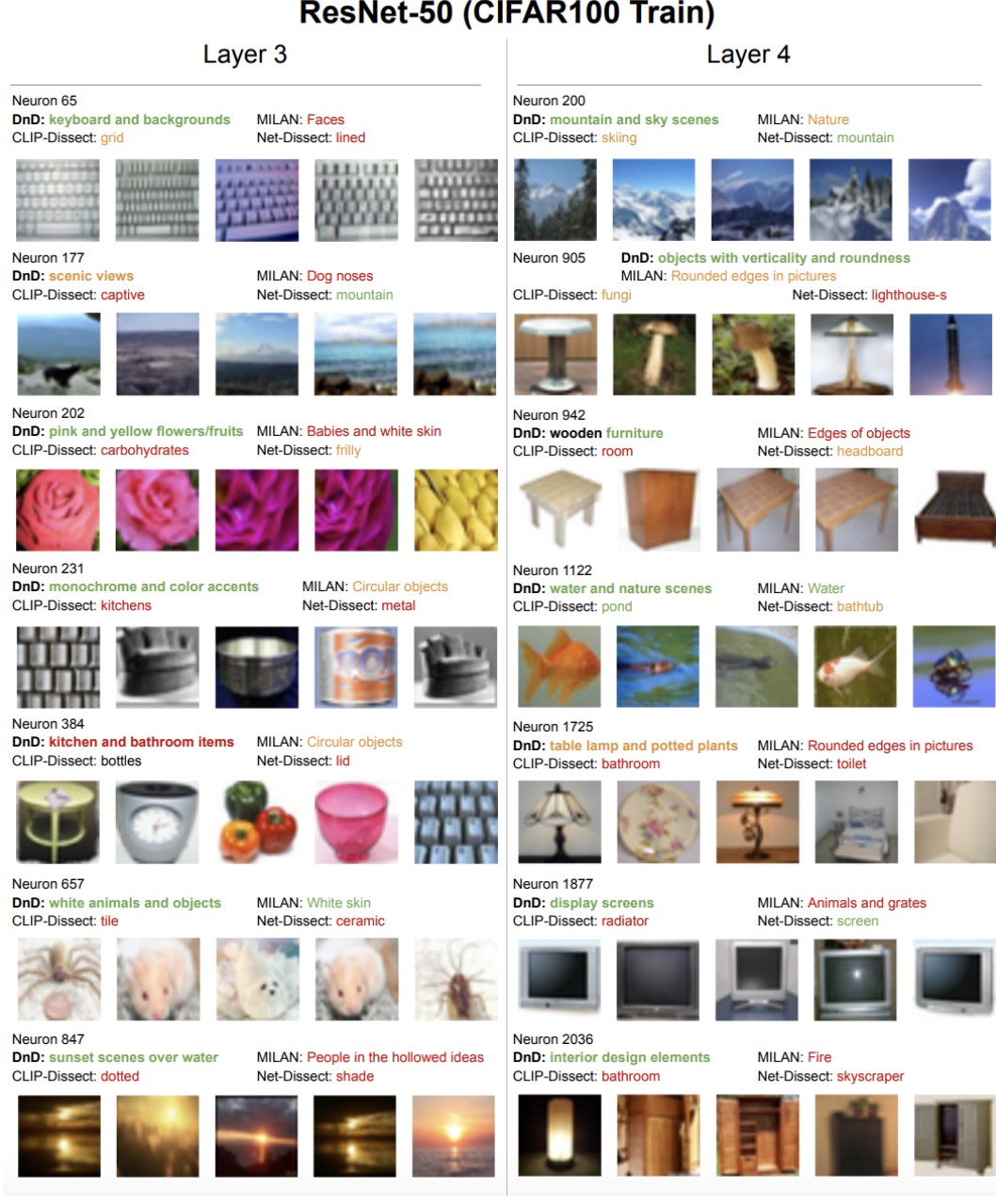

Figure 22: **Examples of DnD results from Layer 3 and 4 of ResNet-50 using the training dataset of CIFAR100 as the probing image set**. We showcase a set of randomly selected neurons and their descriptions from Layer 3 and 4 of ResNet-50 trained on ImageNet-1K. **DnD** is probing dataset agnostic and can maintain its high performance on other image sets. Labels are color-coded by whether we believed they were accurate, somewhat correct/vague or imprecise.

