# OpenReview forum: "Describe-and-Dissect: Interpreting Neurons in Vision Networks with Language Models"
_ICLR.cc/2024/Conference — Submitted to ICLR 2024_

### Official Review · Reviewer_orxh · 2023-10-23

**Soundness:** 2 fair
**Presentation:** 3 good
**Contribution:** 2 fair
**Rating:** 3
**Confidence:** 4

**Summary:**

This method introduces a post-hoc interpretability method based on recent advances in Neuron Identification and LLMs. Specifically, the authors aim to bypass the common limitation of a priori set concept sets using a captioning model, while at the same time trying to filter spurious correlations by generating synthetic images based on Stable Diffusion. Compared to other approaches like CLIP-Dissect, the method augments the set of input data by attention cropping to capture both global and local information thus modeling spatial information. Qualitative evaluation studies suggest that the model exhibits significant improvements compared to the alternatives.

**Strengths:**

This method builds upon recent advances in Neuron Identification and specifically CLIP-Dissect. Within the CLIP-Dissect framework, a probing dataset and a predefined concept are considered, aiming to uncover the individual functionality of each neuron. This is performed via a matching of the images-concept similarity vector and an activation summary of each neuron given the probing set. However, this method fails to take into account spatial information, while being restricted to the predefined concept set, usually comprising single word concepts.

On the other hand, Describe and Dissect (DnD) bypasses the first issue using attention crops of the image; this allows for capturing both global and local infomation. For the second issue, the authors introduce an approach based on the concept generation via a caption model and a summarizer based on LLMs. To select the best concepts and deal with spurious correlations in the original probing set, additional synthetic images are generated via Stable Diffusion and the selection is based on the considered scoring function. The authors introduce appropriate scoring functions and assess their impact.

Despite the usage of several different methods, the paper is overall well written and easy to follow. The methods used are clear and the pipeline consistent.

**Weaknesses:**

However I find some issues with this approach:

(i) Even though the pipeline intuitively makes sense, disentangling the contribution and the impact of each module is very difficult. The authors consider three distinct architectures: (i) BLIP, (ii) GPT and (iii) Stable Diffusion. Each one introduces a different bias to the model, rendering the interpretation of the results in various settings a bit demanding. Moreover, there are additional overheads arising from the attention cropping mechanism derived from contours of the most salient regions. With each introduced module, additional parameters and complications are introduced, such as the number of candidate concepts N,  the number  generated images Q, the number of lowest ranking images beta, and the construction of the prompt for the LLM.

(ii) The effect of each one of the components is not addressed in this work. This applies to both the effect of the hyperparameters but also the modules themselves. This is a very important and missing element of the approach. What is the behavior when the attention crops are removed? What if an augmented concept set is used instead of image to text captioning? How impactful are the synthetic images generated via Stable Diffusion?

(iii) Overall the experimental evaluation of the approach is only based on qualitative human studies with no apparent way to quantify the results. In the alternative CLIP-Dissect method, one could use the original labels as the concept set and assess the matching between the true labels and the neuron identification results. This could be performed either via similarity in the clip space (or using other text encoders). Evidently, this is not possible in the case of DnD.

(iv) In this context, the fact that DnD uses an "augmented" concept set renders the comparison not straightforward. It's not clear to me what are the concept sets used for the other methods. Do the authors consider the respective concept sets used in the original publications? For example the concept set for ImageNet for CLIP-Dissect is the one found in the respective code implementation of said paper? Did the authors try using a different more expressive concept set with the other methods and compare the results? One could argue that by augmenting the concept set used in the CLIP-Dissect case, one can get more expressive concepts that could appear more relevant to a human evaluator.

Despite the fact that this constitutes a post hoc interpretability based on at least 4 black-box models (Attention Cropping, Caption Model, LLM, Stable Diffusion), it's still nevertheless an interesting approach. However, the absence of important ablation studies, as well as the inability to assess the performance of the method in a setting that does not solely rely on human studies, obstructs the thorough assessment and impact of the approach.

**Questions:**

Please see the previous section.

---

> ### Author Response · Authors · 2023-11-23
> **Author response to Reviewer orxh (1/3)**
>
> Thank you for the thoughtful review and good suggestions! Below we discuss some concerns you had and how we have addressed them.
>
> **#1 Effect of each part of the pipeline**
>
> Thank you for the suggestion!
>
> We have discussed the effect of Stable Diffusion’s synthetic images in the original draft, please see sec 4.3 and ablation study in Table 5, Fig 4, where we find that on average DnD with the full pipeline is rated 0.13 (on a scale of 5) higher than DnD without Best Concept Selection.
>
> Following your suggestions, we have conducted additional ablation studies on *attention cropping*, *image captioning*, and *GPT summarization* and we describe the results in below #1(a), #1(b) and #1(c) respectively.
>
> **#1(a) Ablation study on “Attention Cropping”**
>
> We perform an ablation study on the effect of attention cropping to DnD performance in Appendix B.2. We qualitatively observe this change in Figure 11, noticing that the primary concepts for layer 3 neuron 475 and layer 2 neuron 467 are brought out more with attention cropping.
>
> **#1(b) Ablation study on “BLIP Image Captioning”**
>
> We also perform studies exploring the effect of BLIP image captioning on our pipeline. In Appendix B.3, we compare the performance of BLIP and BLIP-2 as image captioners. We found that BLIP and BLIP2 generally give similar results as shown in below Table R1, where we compare the cosine similarity scores between the neuron descriptions generated by BLIP and BLIP2 in a ResNet 50 model. For each layer in the model, we compute the mean CLIP cosine similarity between BLIP and BLIP-2 labels for 50 randomly chosen neurons. Similar conceptual ideas between both models are reflected in the high similarity scores.
>
> It is also interesting that we found there are some cases where BLIP2 is too specific in its captions, causing it to miss the bigger concept and underperform compared to BLIP as seen in Figure 13 in Appendix B.3 in the revised draft. We can see that with neuron 248 in layer 1, the concept should clearly be something along the lines of “black” or “dark.” DnD w/ BLIP is able to capture this with its label “low-light scenes” while DnD w/ BLIP-2 gives a worse label of “monochrome timepieces.” We can see that BLIP-2’s captions are trying too hard to see things in these simple images, detecting the color white and objects where there are none.
>
> *Table R1: Mean Cosine Similarity Between BLIP and BLIP-2 Labels.*
>
> | Metric | Layer 1 | Layer 2 | Layer 3 | Layer 4 | All Layers |
> | - | - | - | - | - | - |
> | Cos Similarity | 0.839 | 0.844 | 0.842 | 0.868 | 0.848 |
>
> To further measure the effect of image captioning in our pipeline, we took your suggestion and experimented with replacing image captioning with a fixed concept set in Appendix B.4. We replace BLIP with CLIP, using CLIP to find text in the 20k concept set that are most similar to the highest activating images of a neuron. The text matched with each highly activating image serve as our image captions, which we then process through the rest of our pipeline. Our qualitative results in section B.4 with Figure 14 show that these short, often single-word captions aren’t descriptive enough for GPT-3.5-Turbo to find semantic similarities in. The more descriptive captions with BLIP yield better results with our pipeline. For quantitative results, we gather the labels produced by this pipeline for all neurons in ResNet-50’s FC layer and follow the experiment #2(b) to find the similarity between the labels and ground truths. We compare to our regular DnD pipeline and display the results in Table 9 (Table R2 below). Our original pipeline slightly outperforms DnD w/ CLIP using both CLIP cosine similarity and MPNet cosine similarity. Note that the final FC layer is likely where a fixed concept set would perform best, as its concise and common text elements are more likely to match up with well-defined class ground truths than the often abstract and polysemantic concepts of intermediate layer neurons. As such, we find the difference to be qualitatively larger on hidden layers and as such image captioning is essential for improving performance.
>
> *Table R2: Mean FC Layer Similarity of CLIP Captioning. Utilizing a fixed concept set (t = 20k) to caption activating images via CLIP Radford et al. (2021), we compute the mean cosine similarity across fully connected layers of RN50. Final labels that do not express concepts are given a similarity score of 0. With both CLIP cosine similarity and MPNet cosine similarity, we find the performance of DnD w/ CLIP Captioning is slightly worse than BLIP generative caption.*
> | Metric / Methods | Describe-and-Dissect (Ours) | DnD w/ CLIP-Captioning | % Decline |
> | - | - | - | - |
> | CLIP cos | **0.7598** | 0.7583 | 0.197% |
> | mpnet cos | **0.4588** | 0.4465 | 2.681% |

---

> > ### Author Response · Authors · 2023-11-23
> > **Author response to Reviewer orxh (2/3)**
> >
> > **#1(c) Ablation study on “GPT Summarization”**
> >
> > The last ablation study we consider is for the role GPT-3.5-turbo plays in our pipeline. We discuss this in Appendix B.5 and display our findings with Figure 16. In this study, we take the image captions as our candidate concepts and then use Best Concept Selection to choose the best caption for the neuron. We see with Figure 16 that there are many failure cases for this pipeline. GPT summarization is necessary to eliminate the noise seen in neuron 155 from layer 1 and to capture the concept linking the activating images rather than just the ones prevalent in one of the images as seen in neuron 418 from layer 2.
> >
> > **#2 Quantify results**
> >
> > **#2(a) Only has qualitative human studies and lacking quantitative results**
> >
> > Thank you for the suggestion! We believe that there might be some misunderstanding here. The human study results are usually considered as “quantitative” analysis in the prior work in this field [1, table 5][2, table 3][3, table 1, 2, A.2]. For example, human evaluators give “quantitative” scores (e.g. ranging from strongly disagree to strongly agree as score 1 to 5) to evaluate the quality of neuron descriptions. Our original evaluation methods followed prior works in this field [1, 2, 3] by utilizing human evaluation for results on the intermediate layers of a network, and we reported the evaluation scores in Table 2, 3, 6, 7. Our results show that we are rated 0.94 better than the state-of-the-art by 37.56%, supporting the effectiveness of our proposed approach.
> >
> > **#2(b) Additional quantitative results**
> >
> > To provide additional quantitative results, we follow [1, e.g. sec 4.2] to evaluate the description accuracy of the final layer neuron. In [1], the authors proposed this idea as an alternative analysis to automatically quantify the neuron description qualities because the “ground-truth” description is available for the final layers (i.e. the class name).
> >
> > We would like to clarify that **it is still possible** to compare DnD quantitatively in this way using embedding similarity. Here, we focus on comparing our method DnD to MILAN [4], as MILAN is the only other contemporary work that provide generative descriptions (see our table 1), while other works e.g. [1, 2] use concept sets which can give them an advantage when calculating the similarity between labels and ground truths. The reason is because they can include the “ground truth” class label or close to it in the concept set. Therefore, it is expected that this evaluation method will be less favorable for generative methods like DnD and MILAN, and it would be more fair to compare DnD and MILAN only. We see this in CLIP-Dissect’s high performance in this experiment [1, sec 4.2] with an average CLIP cosine similarity of 0.7900 and an average MPNet cosine similarity of 0.5257. CLIP-Dissect was also optimized for this task, with its hyperparameters tuned to maximize performance on final layer neurons of ResNet-50.
> >
> > Accordingly, we report our result of a ResNet-50 model trained on ImageNet below in Table R3 and also in Appendix Sec B.6 Table 10 of the revised draft. We use two embeddings (CLIP and MPNet) to determine similarity between the generated neuron descriptions and the ground truth class label. It can be seen that our DnD labels are closer to the ground truths than MILAN’s by a significant margin, indicating the effectiveness of our proposed method.
> >
> > *Table R3: Cosine similarity between predicted labels and ResNet-50’s “ground truths.” We can see that on average, DnD provides labels that are more similar to the ground truths than MILAN’s labels.*
> >
> > | Metric / Methods | MILAN | Describe-and-Dissect (Ours) |
> > | - | - | - |
> > | CLIP cos | 0.7080 | **0.7598** |
> > | mpnet cos | 0.2788 | **0.4588** |
> >
> > **#2(c) Additional Quantitative results: MILANNOTATIONS**
> >
> > We also experimented with using MILANNOTATIONS as another quantitative evaluation but found the annotations to be too noisy to provide a useful signal, to the point that a trivial baseline describing all neurons as “images” outperforms every explanation method. See Appendix B.7 of the revised manuscript for additional details.

---

> ### Author Response · Authors · 2023-11-23
> **Author response to Reviewer orxh (3/3)**
>
> **#2(d) Application: Finding Classifiers for New Concepts**
>
> Finally, we provide a novel application of neuron descriptions to find classifiers for unseen tasks in Appendix B.8, showing we can find relatively good classifiers for unseen tasks. To find a neuron to serve as a classifier, we found the neuron whose description was closest to the CIFAR class name in a text embedding space. Using DnD descriptions we found classifiers with average AUROC of 0.7375 on CIFAR10 classes, and 0.7606 on CIFAR100 classes. In comparison, using MILAN descriptions the average AUROC was only 0.5906 for CIFAR10 and 0.6514 for CIFAR100, likely due to their more generic descriptions. While this is mostly independent as an application of our method it also serves as another quantitative metric showcasing our descriptions are more useful than descriptions generated by alternative methods.
>
> [1] Oikarinen & Weng, Clip-dissect: Automatic description of neuron representations in deep vision networks, 2023
>
> [2] Bau et al., Network dissection: Quantifying interpretability of deep visual representations, 2017
>
> [3] Kalibhat et al., Identifying interpretable subspaces in image representations, 2023
>
> [4] Hernandez et al., Natural language descriptions of deep visual features, 2022
>
>
> **#3 Clarification regarding the role of concept sets in DnD and related works**
>
> One thing we wanted to clarify is that DnD does not utilize a sort of "augmented" concept set. In fact, our method is entirely concept set free. Instead, we generate image captions using BLIP and semantically combine these captions into neuron labels using GPT-3.5-Turbo. In regards to the respective concept sets used in the original publications, we utilize those discussed in their respective papers. For CLIP-Dissect, we employ the 20k concept set as that is the most expansive one discussed in its respective paper which gives it the highest performance possible on intermediate layers of networks. Network Dissection only utilizes the Broden labels which is reflected in our experiments, and MILAN, like DnD, is generative and thus does not employ a concept set.
>
> **#4 Summary**
> In summary we have:
> - **#1** Performed further ablation studies for all components of our pipeline to explore the impact of each component
> - **#2** Added quantitative results through our experiment on the fully-connected layer of ResNet-50, as well as two other qualitative evaluations
> - **#3** Clarified the role concept sets play in our method and other baseline works.
>
> We believe that we have addressed all your concerns. Please let us know if you still have any reservations and we would be happy to address them!

---

### Official Review · Reviewer_pUSu · 2023-10-30

**Soundness:** 3 good
**Presentation:** 3 good
**Contribution:** 2 fair
**Rating:** 6
**Confidence:** 4

**Summary:**

This paper introduces Describe-and-Dissect (DnD), a novel method for interpreting hidden neurons in vision networks. DnD stands out from existing neuron interpretation methodologies by eliminating the need for labeled training data or predefined concept sets, which enhances its broad applicability and generalization capabilities. The approach comprises three primary stages: 1) Enriching the probing image set with crops to encompass both global and local concepts; 2) Utilizing BLIP for generating natural language descriptions of highly activating images, followed by GPT-3.5 to condense these into candidate concepts; 3) Creating new images based on these candidates using Stable Diffusion, and re-evaluating the concepts via a scoring function that considers both original and generated images. The effectiveness of DnD is demonstrated through its application to ResNet-50 trained on ImageNet-1K and ResNet-18 trained on Places365, showcasing superior performance in human evaluations compared to established methods like CLIP-Dissect, MILAN, and Network Dissection.

**Strengths:**

- The research addresses a crucial aspect of model interpretability by enabling natural language interpretation of neurons.

- The proposed method DnD overcomes critical limitations of prior techniques, notably the dependency on large-scale annotated training data or a pre-defined concept bank. Its training-free nature, derived from the integration of various existing foundation models, ensures robust generalization across diverse neural network architectures and data distributions.

- Human evaluations verify the method's effectiveness. DnD's interpretation for ResNet models trained on ImageNet and Places365 was preferred 2x and 3x times by humans compared to prior methods.

- The paper is well-written and easy to understand. The paper has a clear logical flow and detailed experimental results.

**Weaknesses:**

- The approach depends on existing pre-trained foundation models. A more profound exploration of how different model choices impact outcomes would enhance the paper's depth. For instance, the efficacy of DnD highly relies on the captioning technique, as it derives candidate concepts from captions of top activating images. BLIP is used in this work, but its brief captions might limit the method. A comparative analysis with improved captioning systems like BLIP2 or dense captioning models such as LLAVA with prompting, could offer valuable insights into the method's current limitations and potential enhancements.

- The paper mainly relies on human evaluation, making it hard to reproduce the results. Additionally, there is no mention of plans to release the data and human annotation files.

- Section 3.1 on Probing Set Augmentations needs more detail and sufficient ablation studies. The specifics of how attention crops are implemented and their contribution to enhancing DnD's performance remain unclear. Furthermore, Section 4.3, including Figure 4, needs more elaboration on how the synthetic images from Stable Diffusion and the scoring functions enhance DnD's effectiveness.

**Questions:**

- In Table 4, the average human-annotated score for "Top K Squared + Image Products" (representing DnD's final design) is 3.33, notably lower than the 4.15 average in Table 2 and Table 3. Could you clarify the reason for this discrepancy?

- Are there any plans to release the datasets and results for broader access and verification?

---

> ### Author Response · Authors · 2023-11-23
> **Author response to Reviewer pUSu (1/3)**
>
> Thank you for the positive feedback and comments! We would like to address your questions and concerns below.
>
> **#1 Impact of captioning model**
>
> Thank you for the suggestion! Following your suggestions, we have conducted additional experiments using BLIP2 as a captioning model in our pipeline in place of BLIP.
>
> We found that BLIP and BLIP2 generally give similar results as shown in below Table R1, where we compare the cosine similarity scores between the neuron descriptions generated by BLIP and BLIP2 in a ResNet 50 model. For each layer in the model, we compute the mean CLIP cosine similarity between BLIP and BLIP-2 labels for 50 randomly chosen neurons. Similar conceptual ideas between both models are reflected in the high similarity scores.
>
> It is also interesting that we found there are some cases where BLIP2 is too specific in its captions, causing it to miss the bigger concept and underperform compared to BLIP as seen in Figure 13 in Appendix B.3 in the revised draft. We can see that with neuron 248 in layer 1, the concept should clearly be something along the lines of “black” or “dark.” DnD w/ BLIP is able to capture this with its label “low-light scenes” while DnD w/ BLIP-2 gives a worse label of “monochrome timepieces”. We can see that BLIP-2’s captions are trying too hard to see things in these simple images, detecting the color white and objects where there are none.
>
> *Table R1: Mean Cosine Similarity Between BLIP and BLIP-2 Labels.*
> | Metric             | Layer 1     | Layer 2     | Layer 3     | Layer 4     | All Layers     |
> |-----------------|-------------|-------------|-------------|-------------|---------------|
> | Cos similarity | 0.839        | 0.844        | 0.842        | 0.868        | 0.848            |
>
> **#2 Additional Ablation Studies**
>
> In addition to the effects of Image Captioning model studied above, we have conducted several additional ablation studies to study the importance of different parts in our pipeline, summarized below:
> * **Ablation: Attention Cropping.** We added section B.2 in the Appendix where we evaluate the effect of removing attention crops from our pipeline.
> * **Ablation: Fixed Concept Set.** In Appendix B.4 we evaluate how well our network performs if we use CLIP with a fixed concept set instead of a generative image captioning model, showcasing generative models that lead to much improved performance.
> * **Ablation: Effects of GPT Summarization.**  In Appendix B.5 we evaluated the performance of our method if we remove the GPT summarization step, showcasing it leads to noticeably worse concepts.
>
> **#3 Only human evaluation**
>
> Thank you for the suggestion. While we believe that in the end best evaluations of interpretability requires human in the loop, we have provided 3 additional quantitative experiments that are less human dependent.
>
> **#3(a) Additional Quantitative results: Final layer.**
>
> To provide additional quantitative results, we follow [1, e.g. sec 4.2] to evaluate the description accuracy of the final layer neuron. In [1], the authors proposed the idea as an alternative analysis to automatically quantify the quality of neuron descriptions because the “ground-truth” description is available for the final layers (i.e. the class name).
>
> Here, we focus on comparing our method DnD to MILAN [4], as MILAN is the only other contemporary work that provides generative descriptions (see our table 1). Other works e.g. [1, 2] use fixed concept sets which give them an advantage when calculating the similarity between labels and ground truths. This is because the “ground truth” class labels or similar can be incorporated into the concept set. Therefore, it is expected that this evaluation method will be less favorable for generative methods like DnD and MILAN, and would be more fair to compare DnD and MILAN only.
>
> We report our result of a ResNet-50 model trained on ImageNet below in Table R2 and also in Appendix Sec B.6 Table 10 of the revised draft. We use two embeddings (CLIP and MPNet) to determine similarity between the generated neuron descriptions and the ground truth class label. It can be seen that our DnD labels are closer to the ground truths than MILAN’s by a significant margin, indicating the effectiveness of our proposed method.
>
> *Table R2: Cosine similarity between predicted labels and ResNet-50’s “ground truths.” We can see that on average, DnD provides labels that are more similar to the ground truths than MILAN’s labels.*
> | Metric / Methods | MILAN     | Describe-and-Dissect (Ours) |
> |---------------------|------------|-----------------------------------|
> | CLIP cos              | 0.7080     | **0.7598**                                    |
> | mpnet cos           | 0.2788     | **0.4588**                                     |

---

> ### Author Response · Authors · 2023-11-23
> **Author response to Reviewer pUSu (2/3)**
>
> **#3(b) Additional Quantitative results: MILANNOTATIONS**
>
> We also experimented with using MILANNOTATIONS as another quantitative evaluation but found the annotations to be too noisy to provide a useful signal, to the point that a trivial baseline describing all neurons as “images” outperforms every explanation method. See Appendix B.7 of the revised manuscript for additional details.
>
> **#3(c) Application: Finding Classifiers for New Concepts**
>
> Finally we provide a novel application of neuron descriptions to find classifiers for unseen tasks in Appendix B.8, showing we can find relatively good classifiers for unseen tasks. To find a neuron to serve as a classifier, we found the neuron whose description was closest to the CIFAR class name in a text embedding space. Using DnD descriptions we found classifiers with average AUROC of 0.7375 on CIFAR10 classes, and 0.7606 on CIFAR100 classes. In comparison, using MILAN descriptions the average AUROC was only 0.5906 for CIFAR10 and 0.6514 for CIFAR100, likely due to their more generic descriptions. While this is mostly independent as an application of our method it also serves as another quantitative metric showcasing our descriptions are more useful than descriptions generated by alternative methods.
>
> [1] Oikarinen & Weng, Clip-dissect: Automatic description of neuron representations in deep vision networks, 2023
>
> [2] Bau et al., Network dissection: Quantifying interpretability of deep visual representations, 2017
>
> [3] Kalibhat et al., Identifying interpretable subspaces in image representations, 2023
>
> [4] Hernandez et al., Natural language descriptions of deep visual features, 2022
>
> **#4 Code/data release**
>
> Thank you for the question, we are committed to release the source code, data and human evaluation results in the camera-ready version for reproducibility of our result.
>
> **#5 Attention Cropping Details (Sec 3.1)**
>
> Following your suggestion, we provided more details on Sec 3.1, Sec 4.3 and Fig 4 below and in the revised draft.
>
> Thanks for pointing this out, following your suggestion we have included a detailed description of attention cropping in Appendix B.1 of the revised draft, including Figure 10 to visualize the procedure.
>
> We also follow your suggestion to do an ablation study on the effect of attention cropping to DnD performance in Appendix B.2. We qualitatively observe this change in Figure 11, noticing that the primary concepts for layer 3 neuron 475 and layer 2 neuron 467 are brought out more with attention cropping.
>
> **#6 Additional details on Concept Selection (Sec 4.3, Fig. 4)**
>
> To expand more on section 4.3, we elaborate on the importance of Best Concept Selection. Due to GPT-3.5-Turbo’s generative and variable nature, it can produce different outputs each time when given the same input. This means that if you simply generate one concept label from it for a neuron, it can give you different labels each time; obviously some of these labels will be better than others. To account for this fact and maximize performance, we employ Best Concept Selection. Generating images from the multiple labels GPT can produce for one neuron, we calculate which label activates the neuron the most and is thus most likely to represent what the neuron encodes. Generating new images helps us remove spurious concepts that can be present in the highly activating images of a neuron due to correlation, but do not by themselves cause the neuron to fire. An example of this is in Figure 4b, where DnD without Best Concept Selection gives “Monochromatic and patterned designs.” We can see that monochromatic is a spurious concept not required for high activation of neuron 927, and as such DnD with Best Concept Selection accounts for this and labels the neuron “abstract swirls.” This is far closer to the true label of the neuron and signifies why Best Concept Selection is essential for our pipeline to give its best performance.

---

> ### Author Response · Authors · 2023-11-23
> **Author response to Reviewer pUSu (3/3)**
>
> **#7 Clarification on discrepancy between Table 4 v.s. Table 2**
>
> Thank you for asking this question!
>
> For experiment 4.2, we augment the DnD pipeline by adding Network Dissection, MILAN, and CLIP-dissect labels along with step 2's candidate concepts into the set of potential concepts fed into Best Concept Selection to increase diversity in potential candidate concepts. The purpose of this experiment is to find the highest performing scoring function, which we accomplished through Table 4 and deemed “TopK Squared + Image Products” to be the highest performing scoring function. By adding more diverse potential labels into the mix, we aimed to enhance the difference in quality between each scoring function’s selected label. Even then, we found that including descriptions from other methods into candidate concepts decreased average concept quality, as the scoring function may still occasionally select a worse label from the other methods, causing its average evaluation score to not be as high.
>
> However, we can see that “TopK Squared + Image Products” in Table 4 and Describe-and-Dissect in Table 6 follow the same general pattern in score when traversing the intermediate layers of ResNet-50: the score starts high in layer 1, drops a bit in layer 2, and then gradually increases until layer 4. As such we can see that DnD is still generally performing the same in both experiments.
>
> Additionally, these experiments were performed with two different groups of evaluators, so the score ranges are also naturally different. To exemplify this point, we note that the evaluators for Tables 4 and 5 are the same evaluation group, but different from the one in Table 2. We can see that the evaluation group from Tables 4 and 5 generally score all labels lower than the one from Table 2, as the former group gave an average rating of 3.97 to DnD’s labels while the latter group gave an average rating of 4.15. Though these scores are more similar, this difference alongside the aforementioned augmentation of the DnD pipeline for Table 4 explain the discrepancies between the scores for Table 2 and Table 4. As such we don’t intend to compare values between tables, only within the same table. We will make this point clear in the draft.
>
> **#8 Summary**
>
> In summary we have:
> * **#1** Added an analysis on using a different captioning model (BLIP2)
> * **#2** Added several other ablation studies to understand the importance of different parts of our pipeline
> * **#3** Provided 3 additional experiments not reliant on human evaluation
> * **#4** Clarified that we plan to release our code and data with the camera-ready version
> * **#5** Clarified attention cropping pipeline and its effects
> * **#6** Discussed the importance of using synthetic images, and scoring functions
> * **#7** Clarified the discrepancy in scores between Table 4 and Tables 2
>
> Please feel free to let us know if you have any additional questions or comments, and we would be happy to discuss further!

---

### Official Review · Reviewer_gXLM · 2023-10-30

**Soundness:** 3 good
**Presentation:** 2 fair
**Contribution:** 2 fair
**Rating:** 5
**Confidence:** 4

**Summary:**

The paper proposes Describe-and-Dissect, a new method to describe neurons of a neural network with natural language descriptions. The multi-stage procedure first observes the activation of a target neuron for a set of images and image crops of higher activated regions, then captions top activating images with BLIP and summarizes the generated text into a list of concept proposals with GPT-3.5. Finally, a text-to-image generation model is conditioned on the concept proposals to generate images which are used to obtain neuron activations and ultimately a score to select the best matching concept. The neuron descriptions generated by Describe-and-Dissect are preferred over existing methods based on human evaluations.

**Strengths:**

- Describe-and-Dissect does not require training data and generates open-set concepts without human input or guidance.
- The pipeline is reasonable, utilizing foundation models where appropriate, and its components are well explained.
- The user study underlines the efficacy of the method compared to existing methods.
- Ablation studies justify design choices to the most extend, i.e., the scoring function and using a generative model to score the concepts.

**Weaknesses:**

- While there has been precedence of other methods trying to interpret neurons and when they activate, the motivation of this problem is rather weak without a clear application. MILAN showcased an application where neurons were suppressed in order to remove spurious correlations. It would have helped the presentation of the paper if such an application was evaluated and compared to competitors.
- At least a subset of neurons are considered to be polysemantic [1-7], that is, they do not activate on a single concept but rather activate on multiple seemingly unrelated concepts. This issue is not discussed at all in the paper and the proposed method makes the assumption that a single concept exists as step 3 filters a single concept from a list of candidate concepts. A recent method [8] instead tries to decode full representations of layers instead of individual neurons. At least some of these related works should be discussed, but ideally the issue of polysemanticity could have been analyzed as a potential limitation/failure case of the model.
- Related to the previous point, it is unclear what the proposed method predicts for neurons that are unexplainable (e.g. due to polysemanticity or when the neuron activates on concepts not understandable by humans). How often does it happen that each of the candidate concepts describes a subset of the images the neuron activates on, but not a single concept describes all images? One limitation seems to be that in case of polysemantic neurons, Describe-and-Dissect can only extract one of the concepts and this should be clarified and possibly even evaluated. Qualitative examples of failure cases could make the picture more complete.
- The crowdsourced experiment did not indicate the location of the neuron activation in each image. Based on the MILAN dataset it seems crucial to have this information to determine the semantic concept the neuron activates on. Why was this choice made? This type of evaluation could create a bias towards explanation methods that describe global image content. Since a captioning model was used (BLIP), part of the performance in the user study could be explained by this image captioning bias. The authors could have also used the MILAN dataset for evaluation, but this experiment is missing.
- The authors mention that human raters have deemed some neurons to be unexplainable (caption of Table 4). This begs the question how we could tell whether a description of a neuron is trustworthy (neuron is easily explainable/monosemantic) or whether we should not rely on the generated description (neuron is not interpretable) without asking humans about every neuron.

[1] Olah et al., Feature Visualization, Distill, 2017
[2] Olah et al., Zoom In: An Introduction to Circuits, Distill, 2020
[3] Mu et al., Compositional Explanations of Neurons, NeurIPS 2020
[4] O'Mahony et al., Disentangling Neuron Representations with Concept Vectors, CVPR Workshops, 2023
[5] Scherlis et al., Polysemanticity and Capacity in Neural Networks, arxiv, 2022
[6] Gurnee et al., Finding Neurons in a Haystack: Case Studies with Sparse Probing, arxiv, 2023
[7] Bricken et al., Towards Monosemanticity: Decomposing Language Models With Dictionary Learning, Transformer Circuits Thread, 2023
[8] Dani et al., DeViL: Decoding Vision features into Language, GCPR, 2023

**Questions:**

- All hyperparameters should be listed for reproducibility purposes, i.e. $\alpha$ and $K$ (images used in step 1) do not seem to be specified.
- What is the influence of important hyperparameters, e.g., $\alpha$, $K$, $N$, $Q$? Why were the reported values chosen?
- What function is used for $g$? Do you spatially average pool the activations of a neuron?
- In Table 4, the captions mentions "neurons deemed uninterpretable by raters were excluded". How was this determined by the workers? Why was this data excluded? Apart from this not being too relevant for the comparison of scoring functions, this data is highly relevant and interesting for the overall evaluation of the method.

---

> ### Author Response · Authors · 2023-11-23
> **Author response to Reviewer gXLM (1/3)**
>
> Thank you for the thoughtful review and good suggestions! Below we discuss some concerns you had and how we have addressed them.
>
> **#1 Motivation/application**
>
> Thank you for the suggestion!
>
> While we were unable to reproduce the spurious correlation experiment of MILAN because they have not released the dataset used for the experiment, we came up with a new application to showcase the usefulness of our neuron descriptions.
>
> To showcase a potential use case for neuron descriptions (and provide another way to quantitatively compare explanation methods), we experimented with using neuron descriptions to find a good classifier for a class missing from the training set.
>
> Our setup is as follows: we explained all neurons in layer4 of ResNet-50(ImageNet) using different methods. We then wanted to find neurons in this layer that could serve as the best classifiers for an unseen class, specifically the classes in CIFAR-10 and CIFAR-100 datasets. Note there is some overlap between these and ImageNet classes, but CIFAR classes are typically much more broad. To find a neuron to serve as a classifier, we found the neuron whose description was closest to the CIFAR class name in a text embedding space (ensemble of CLIP ViT-B/16 text encoder and MPNet text encoders used in this paper). We then measured how well that neuron(its average activation) performs as a single class classifier on the CIFAR validation dataset, measured by area under ROC curve. For cases where multiple neurons shared the closest description, we averaged the performance of all neurons with that description. Results are shown below in Table R1:
>
> *Table R1: The average classification AUC on out of distribution dataset when using neurons with similar description as a classifier. We can see Describe-and-Dissect clearly outperforms MILAN, the only other generative description method.*
>
> | Metric / Methods | MILAN | Describe-and-Dissect (Ours) |
> | ------------------- | --------- | -------------------------------- |
> | CLIP cos | 0.5906 | **0.7375** |
> | mpnet cos | 0.6516 | **0.7607** |
>
> We can see DnD performs quite well, reaching AUROC values around 0.75, while MILAN performs much worse. We believe this may be because MILAN descriptions are very generic (likely caused by noisy dataset, see Appendix B.7), which makes it hard to find a classifier for a specific class. We think this is a good measure of explanation quality, as different methods are dissecting the same network, and even if no neurons exist that can directly detect a class, a better method should find a closer approximation.
>
> We have included this discussion in Appendix B.8 of the revised manuscript.
>
> **#2 Address polysemanticity**
>
> Thank you for the suggestions!
>
> We have added the following discussion on polysemanticity under Appendix A.1: Limitations:
>
> **Polysemantic neurons:** Existing works Olah et al. (2020); Mu & Andreas (2020); Scherlis et al. (2023) have shown that many neurons in common neural networks are polysemantic, i.e. represent several unrelated concepts or no clear concept at all. This is a challenge when attempting to provide simple text descriptions to individual neurons, and is a limitation of our approach, but can be some- what addressed via methods such as adjusting DnD to provide multiple descriptions per neuron as we have done in the Appendix B.9. Another way to address this is by providing an interpretability score, which can differentiate good descriptions from poor ones, which we have explored in the Appendix B.10. We also note that due to the generative nature of DnD, even its single labels can often encapsulate multiple concepts by using coordinating conjunctions and lengthier descriptions. However, polysemantic neurons still remain a problem to us and other existing methods such as Bau et al. (2017) Hernandez et al. (2022) Oikarinen & Weng (2023). One promising recent direction to alleviate polysemanticity is via sparse autoencoders as explored by Bricken et al. (2023)
>
> To address this issue of polysemanticity, we have performed an experiment in Appendix B.9 in which we modify the DnD pipeline to provide multiple neuron labels when necessary, rather than just filtering for a single concept. The scoring function now chooses the top 3 candidate labels rather than just the best one. If the similarity computed by CLIP between two of these labels exceeded a certain threshold (for our purposes we used a threshold of 0.81), only the top label would be taken and the other would be discarded. Through qualitative analysis in section Figure 18, we show that this method is capable of capturing multiple concepts of a neuron. With neuron 508 from layer 2, DnD is able to capture not only that the neuron encodes for polka dot and metal textures, but also that the textures are primarily black and white. Similarly with neuron 511 from layer 3, DnD labels the neuron as not only primarily encoding for interior elements, but also finds that these elements are black and white.

---

> > ### Author Response · Authors · 2023-11-23
> > **Author response to Reviewer gXLM (2/3)**
> >
> > **#3 Uninterpretable Neurons**
> >
> > Additionally, we want to clarify how DnD deals with unexplainable neurons. When faced with one, DnD tries its best to find a common factor between the image captions. Sometimes it might focus on a concept only shared by some of the images, giving its best guess, or provide a very broad label such as “patterns” to try covering everything. Examples of this can be seen in Figure 19a, as with neuron 525 in layer 3 it describes only a subset of the highest activating images with its label “tools and renewable energy” and with neuron 308 in layer 3 it provides a very broad label of “different activities or objects.”
> >
> > To address your point about how we can tell a description of a neuron is trustworthy, we developed an “interpretability score” for neurons in Appendix B.10. We did this by taking the mean similarity between the embeddings for a neuron’s top activating images. As seen in Figure 19, we can use this score to gauge how interpretable a neuron is and thus how confident we are in DnD’s ability to describe it.
> >
> > Finally, we want to address your questions regarding how we dealt with uninterpretable neurons in Table 4 and the Mechanical Turk results in section 4.2. For Table 4, the workers would look at the top ten activating images of the target neuron and if they deemed that there was likely no possible label that could encapsulate most of the images, they would record the neuron as uninterpretable. The data for uninterpretable neurons was included in the overall comparison/evaluation with other methods in Tables 2 and 3, and workers on those tasks did not have the option to label a neuron as uninterpretable. It was only excluded from the experiment in Table 4 as that experiment’s purpose was to determine the scoring function best at selecting labels for interpretable neurons, and adding uninterpretable neurons into the mix would make the results less reliable.
> >
> > **#4 Activation Highlighting**
> >
> > Our crowdsourced experiment did indicate the location of the neuron activation in each image through our inclusion of attention cropping in DnD’s pipeline (see 3.2, B.1). The images presented in the Mechanical Turk experiments were results of this attention cropping, and as such the images shown were the regions that the neuron was detecting. While this is different from how activations are highlighted in MILAN, it is not clear to us if one method is better than another, or which should be used for evaluation. In light of these challenges, we have added a section in Appendix A.1. Limitations discussing the challenges with providing a fair comparison between methods.
> >
> > **#5 Using MILANNOTATIONS**
> >
> > Per your suggestion, we added an experiment on the MILAN dataset in Appendix B.7. We dissected the first four layers of ResNet-152 and used CLIP and MPNet to calculate the similarity between methods’ labels and the “ground truth” MILANNOTATIONS. We compare against MILAN and CLIP-Dissect in Table 11. We only look at random “reliable” neurons, which are defined as having at least two corresponding MILANNOTATIONS whose CLIP cosine similarity exceeds 0.81. Our results are generally mixed, with MILAN performing the best (average of 0.7698), then DnD (average of 0.7543), and finally CLIP-Dissect (average of 0.7390) using CLIP similarity. However, with MPNet similarity, CLIP-Dissect is calculated as the best (average of 0.1970) followed by MILAN (average of 0.1391) and then DnD (average of 0.1029).
> >
> > However, we found this experiment to be unreliable/not useful when comparing methods, as the MILANNOTATIONS dataset is very noisy with the multiple annotations per neuron often being unrelated, as seen in Table 11 (average CLIP similarity between annotations of the same neuron is 0.7884, note that this result is significantly higher than the whole dataset as we only look at “reliable” neurons). Out of 200 neurons across 4 layers of ResNet-152, only 103 have at least 2 MILANNOTATIONS whose CLIP cosine similarity exceeds 0.81. Even further, only 14 of those have all three MILANNOTATIONS whose CLIP cosine similarities with each other exceed 0.81. Figure 9 qualitatively depicts how variable the annotations within each neuron can be, and we can see this for example in layer 4 neuron 883. This neuron’s MILANNOTATIONS are “Circular items,” “animals and vehicles,” and “the area under an object.” Each of these annotations are very different from each other and, looking at the highest activating images, don’t describe the neuron that well.
> >
> > To further showcase the issues caused by this, we hypothesized that the noisiness of annotations favors very generic descriptions as they are relatively close to everything, which turned out to be the case, as a trivial baseline of describing every neuron as “Images” noticeably outperforms all 3 description methods on all layers, showcasing that high score on MILANNOTATIONS does not indicate the descriptions are good.

---

> ### Author Response · Authors · 2023-11-23
> **Author response to Reviewer gXLM (3/3)**
>
> **#6 Hyperparameters**
>
> Following your suggestion, we have gone through in the revised draft and defined what values we set certain hyperparameters to if we didn’t already in the original draft.
>
> This paragraph is dedicated to addressing your question regarding the influence of certain hyperparameters and our reasoning behind setting them to the values we did.
> - $\alpha$ was set to 4 as after some testing we found that value to be reasonable. Less than that and we could miss out on important concepts in an image that are causing the neuron to activate; more than that and our top activating images could be dominated by crops of one image, preventing us from taking the other highly activating images into account.
> - $K$ was set to 10 as we found through testing that a value lower than this could exclude valuable information in other highly activating images and a value higher than this could introduce spurious concepts that confuse GPT.
> - $N$ was set to 5 as we observed that GPT rarely produced over 5 distinct candidate concepts, so increasing $N$ beyond that would be superfluous. Decreasing $N$ below 5 could possibly eliminate accurate potential labels from being fed into Best Concept Selection.
> - $Q$ was set to 10 to balance quality with runtime/computational cost, as while the quality of Best Concept Selection increases with $Q$, so does the runtime.
> - $\eta$ was set to 0.5 as, intuitively, it makes sense for at least the majority of the space covered in our bounding boxes to be distinct; otherwise the attention crops would all be very similar. If $\eta$ was less than 0.5, we could find a problem of the crops being too small, preventing the image captioner from perceiving comprehensive concepts.
>
> Finally, we clarified in the revised draft that, unless specified, the summary function g we use for our experiments is spatial mean. Spatial mean is used as the summary function in all experiments except for the MILANNOTATIONS experiment in section B.7 and Table 11. This experiment uses spatial max as the summary function since the “ground truth” MILANNOTATIONS were created on the highest activating images calculated with max pooling.
>
> **#7 Summary**
>
> In summary we have:
> - **#1** Added an application/use-case of DnD
> - **#2** Performed an experiment in which DnD provides multiple labels to address issues of polysemanticity and added a discussion of polysemanticity to our limitations
> - **#3** Developed a method for finding which neurons are uninterpretable and clarified our methodology regarding uninterpretable neurons.
> - **#4** Discussed crowdsourced evaluation choices and the difficulty of providing a fair evaluation between methods
> - **#5** Performed an experiment comparing our labels to MILANNOTATIONS, and shown why this is not a useful measure of description quality
> - **#6** Clarified our hyperparameters and what summary function g we used
>
> We believe that we have addressed all your concerns. Please let us know if you still have any reservations and we would be happy to address them!

---

### Official Review · Reviewer_YsCF · 2023-11-01

**Soundness:** 3 good
**Presentation:** 2 fair
**Contribution:** 2 fair
**Rating:** 5
**Confidence:** 3

**Summary:**

This paper utilizes a captioning model and a large language model to provide descriptions for highly activated images for corresponding neurons.

**Strengths:**

- The Describe-and-Dissect method presented in this paper addresses the limitations of previous research and is a model-agnostic, training-free approach.
- The authors provide adequate ablation studies to study the difference between their chosen scoring function and conception selection.
- The authors demonstrate that their proposed approach is qualitatively better than the baseline methods through crowdsourced experiments.

**Weaknesses:**

- The authors use crowdsourced experiments to compare with the baseline method, providing qualitative results but lacking quantitative ones, which falls short of demonstrating the effectiveness of the proposed approach.
- The authors only use 2 datasets and 2 models  to check the efficacy of their approach. To make the paper stronger and for completeness, it will be beneficial to include more datasets and models.

**Questions:**

Please address the weaknesses above

---

> ### Author Response · Authors · 2023-11-23
> **Author response to Reviewer YsCF (1/2)**
>
> Thank you for the feedback! Please see our response below to address your concerns.
>
> **#1 Quantitative results**
>
> **#1(a) Only has crowdsourced experiments and lacking quantitative results**
>
> Thank you for the suggestion! We believe that there might be some misunderstanding here. The human study results are usually considered as “quantitative” analysis in the prior work in this field, please see [1, table 5][2, table 3][3, table 1, 2, A.2]. For example, human evaluators give “quantitative” scores (e.g. ranging from strongly disagree to strongly agree as score 1 to 5) to evaluate the quality of neuron descriptions. Our original evaluation methods followed prior works in this field [1, 2, 3] by utilizing human evaluation for results on the intermediate layers of a network, and we reported the evaluation scores in Table 2, 3, 6, 7. Our results show that we are on average rated 0.94 better than the state-of-the-art models (which is a 37.56% increase), supporting the effectiveness of our proposed approach.
>
>
> **#1(b) Additional Quantitative results: Final layer.**
>
> To provide additional quantitative results, we follow [1, e.g. sec 4.2] to evaluate the description accuracy of the final layer neuron. In [1], the authors proposed the idea as an alternative analysis to automatically quantify the quality of neuron descriptions because the “ground-truth” description is available for the final layers (i.e. the class name).
>
> Here, we focus on comparing our method DnD to MILAN [4], as MILAN is the only other contemporary work that provides generative descriptions (see our table 1). Other works e.g. [1, 2] use fixed concept sets which give them an advantage when calculating the similarity between labels and ground truths. This is because the “ground truth” class labels or similar can be incorporated into the concept set. Therefore, it is expected that this evaluation method will be less favorable for generative methods like DnD and MILAN, and would be more fair to compare DnD and MILAN only.
>
> We report our result of a ResNet-50 model trained on ImageNet below in Table R1 and also in Appendix Sec B.6 Table 10 of the revised draft. We use two embeddings (CLIP and MPNet) to determine similarity between the generated neuron descriptions and the ground truth class label. It can be seen that our DnD labels are closer to the ground truths than MILAN’s by a significant margin, indicating the effectiveness of our proposed method.
>
> *Table R1: Cosine similarity between predicted labels and ResNet-50’s “ground truths.” We can see that on average, DnD provides labels that are more similar to the ground truths than MILAN’s labels.*
> | Metric / Methods | MILAN | Describe-and-Dissect (Ours) |
> | -------------------- | -------- | -------------------------------- |
> | CLIP cos | 0.7080 | **0.7598** |
> | mpnet cos | 0.2788 | **0.4588** |
>
>
> **#1(c) Additional Quantitative results: MILANNOTATIONS**
>
> We also experimented with using MILANNOTATIONS as another quantitative evaluation but found the annotations to be too noisy to provide a useful signal, to the point that a trivial baseline describing all neurons as “images” outperforms every explanation method. See Appendix B.7 of the revised manuscript for additional details.
>
>
> **#1(d) Application: Finding Classifiers for New Concepts**
>
> Finally we provide a novel application of neuron descriptions to find classifiers for unseen tasks in Appendix B.8, showing we can find relatively good classifiers for unseen tasks. To find a neuron to serve as a classifier, we found the neuron whose description was closest to the CIFAR class name in a text embedding space. Using DnD descriptions we found classifiers with average AUROC of 0.7375 on CIFAR10 classes, and 0.7606 on CIFAR100 classes. In comparison, using MILAN descriptions the average AUROC was only 0.5906 for CIFAR10 and 0.6514 for CIFAR100, likely due to their more generic descriptions. While this is mostly independent as an application of our method it also serves as another quantitative metric showcasing our descriptions are more useful than descriptions generated by alternative methods.
>
> [1] Oikarinen & Weng, Clip-dissect: Automatic description of neuron representations in deep vision networks, 2023
>
> [2] Bau et al., Network dissection: Quantifying interpretability of deep visual representations, 2017
>
> [3] Kalibhat et al., Identifying interpretable subspaces in image representations, 2023
>
> [4] Hernandez et al., Natural language descriptions of deep visual features, 2022

---

> > ### Author Response · Authors · 2023-11-23
> > **Author response to Reviewer YsCF (2/2)**
> >
> > **#2 New experiments on additional models and dataset**
> >
> > Following your suggestion, we have conducted an experiment on an additional model (ViT-B/16) to show that our method performs well on a multitude of diverse models. Figure 20 in section B.11 displays that DnD maintains its quality of labels on other models.
> >
> > We also display another experiment in Appendix B.11 with figures 21 and 22 on ResNet-50 using a different probing dataset (CIFAR 100 Train) to show that our method is not reliant on specific datasets.
> >
> > For these two experiments’ figures, the labels are color-coded based on whether we deemed them as accurate (green), somewhat correct/vague (yellow), or imprecise (red). We can qualitatively tell from these figures that DnD’s labels are more accurate than other methods.
> >
> >
> > **#3 Summary**
> >
> > In summary we have:
> > - Clarified the quantitative analysis in the literature and performed 3 additional quantitative experiments in **#1** to demonstrate the effectiveness of our proposed method
> > - Performed further experiments in **#2** to include more models and datasets in our results
> >
> > We believe that we have addressed all your concerns. Please let us know if you still have any reservations and we would be happy to address them!

---

### Author Response · Authors · 2023-11-23
**General response: Overview of New Results**

We thank all the reviewers for the constructive feedback and suggestions!

In response to the reviews, we have conducted many new experiments as requested, mostly focused on ablating the importance of different parts of our proposed methods and providing more/broader results to showcase the capability of our method.

All the new results are reported in Appendix B (p.21-p.37) with “blue text” in the section title / heading with the contents in black for clarity. For the changes in the original draft (main page 1-9 & appendix A), we highlighted the text changes with blue colors for clarity.

**New experiments:**

**1. Ablation: Attention Cropping.**
 We added section B.2 in the Appendix where we evaluate the effect of removing attention crops from our pipeline.

**2. Ablation: Image-to-Text model.**
In Appendix B.3 we evaluate the effect of using BLIP-2 instead of BLIP as the image captioning model, and show that our results are not very sensitive to the choice of image captioning model.

**3. Ablation: Image Captioning with Fixed Concept Set.**
 In Appendix B.4 we evaluate how well our network performs if we use CLIP with a fixed concept set instead of a generative image captioning model, showcasing generative models that lead to much improved performance.

**4. Ablation: Effects of GPT Concept Summarization.**
In Appendix B.5 we evaluated the performance of our method if we remove the GPT summarization step, showcasing it leads to noticeably worse concepts.

**5. Quantitative Results: Final layer.**
In Appendix B.6 we evaluate our descriptions on the final layer neurons where we know the ground truth concept activation, showing we outperform the generative descriptions of MILAN by a large margin.

**6. Quantitative Results: MILANNOTATIONS.**
In Appendix B.7, we explore using the MILANNOTATIONS dataset to evaluate hidden layer neuron descriptions. However, we discover that the dataset is too noisy/low quality to provide reliable results, and a trivial baseline describing every neuron as “images” outperforms all methods.

**7. Application: Finding Classifiers for New Concepts.**
In Appendix B.8, we apply the description to find neurons that work well as classifiers for unseen concepts. We show DnD can help us find relatively high quality classifiers, significantly better than MILAN.

**8. Multiple Labels.**
In Appendix B.9, we experiment with returning multiple labels for a neuron when the top descriptions are sufficiently different to help capture polysemantic neurons.

**9. Quantifying Neuron Interpretability.**
We propose a new way to quantify how interpretable a neuron is in Appendix B.10, by measuring the similarity between its highly activating images.

**10. Results on more models and datasets.**
In Appendix B.11, we provide additional qualitative results by dissecting a vision transformer, showing our method works well on completely different model architectures, as well as dissecting RN50 using CIFAR100 as the probing dataset, showing our method works with this dataset as well.


**Changes:**
1. Clarified our description in section 3 to include the hyperparameter choices we used.

2. Added a section discussing polysemanticity and the challenges it poses to Appendix A.1: Limitations, as well as some discussion on strategies to address it as suggested by reviewer gXLM.

3. Added a section describing challenges in running comparisons based on highly activating images between the methods in Appendix A.1: Limitations.
4. Added section B.1 to provide a more detailed description of our attention cropping pipeline.

---

### Meta-Review · Area_Chair_v9Zt · 2023-12-29

**Metareview:**

This paper addresses the challenge of interpretability in deep neural networks for vision. It proposes a method called Describe-and-Dissect, where the role of hidden neurons is described by pretrained multimodal models. Results are evaluated in a human study.

Initial reviews noted a number of strengths and weaknesses. Authors provided a detailed rebuttal, including additional experimental results. With a rebuttal that was submitted relatively late, there was limited author-reviewer discussion. follow-up discussion between reviewers and between reviewers and AC identified several aspects that had been addressed successfully. However, a number of concerns remained, in particular regarding evaluation of the method. Given that the AC-reviewer discussion was not visible to the authors, I summarize main points below to help the authors improve the manuscript.

During the rebuttal, the authors added quantitative evaluation as requested by multiple reviewers. Reviewers remain unconvinced about the new evaluation technique's use of CLIP for assessing textual similarity. A reviewer noted that "previous approaches [1,2] used BERTScore and other NLP evaluation metrics that are more fine-grained". In addition, a reviewer raised concerns regarding the score differences between Table 2/3 and 4. While the authors interpret these as attributable to variance, the reviewer notes that "this suggests a lack of a consistent and clear guideline for annotation, which is a critical issue that needs to be addressed". Further concerns were raised regarding the use of attention cropping and that human judges were shown images resulting from such crops, meaning that they saw different images for different approaches for generating descriptions.

Reviewers and AC commend the important direction of this work, but given the remaining concerns the work cannot be recommended for acceptance in its present state.


[1] Hernandez et al., Natural language descriptions of deep visual features, ICLR 2022
[2] Dani et al., DeViL: Decoding Vision features into Language, GCPR, 2023

**Justification For Why Not Higher Score:**

Too many concerns about evaluation (and quantiative evaluation was a major new addition during the rebuttal phase)

**Justification For Why Not Lower Score:**

N/A

---

### Decision · Program_Chairs · 2024-01-16

Reject